# Temporal evolution of crack propagation propensity in snow in relation to slab and weak layer properties

Jürg Schweizer, Benjamin Reuter, Alec van Herwijnen, Bettina Richter, Johan Gaume

WSL Institute for Snow and Avalanche Research SLF, Flüelastrasse 11, 7260 Davos Dorf, Switzerland

*Correspondence to*: Jürg Schweizer (schweizer@slf.ch)

**Abstract.** If a weak snow layer below a cohesive slab is present in the snow cover, unstable snow conditions can prevail for days or even weeks. We monitored the temporal evolution of a weak layer of faceted crystals as well as the overlaying slab layers at the location of an automatic weather station in the Steintälli field site above Davos (Eastern Swiss Alps). We focussed on the crack propagation propensity and performed propagation saw tests (PSTs) on seven sampling days during a

two-month period from early January to early March 2015. Based on video images taken during the tests we determined the mechanical properties of the slab and the weak layer and compared them to the results derived from concurrently performed measurements of penetration resistance using the snow micro-penetrometer (SMP). The critical cut length, observed in PSTs, overall increased during the measurement period. The increase was not steady and the lowest values of critical cut length were observed around the middle of the measurement period. The relevant mechanical properties, the slab effective elastic

modulus and the weak layer specific fracture, overall increased as well. However, the changes with time differed suggesting that the critical cut length cannot be assessed by simply monitoring a single mechanical property such as slab load, slab modulus or weak layer specific fracture energy. Rather, crack propagation propensity is the result of a complex interplay between the mechanical properties of the slab and the weak layer. We then compared our field observations to newly developed metrics of snow instability related to either failure initiation or crack propagation propensity. The metrics were

either derived from the SMP signal or calculated from simulated snow stratigraphy (SNOWPACK). They partially reproduced the observed temporal evolution of critical cut length and instability test scores. Whereas our unique dataset of quantitative measures of snow instability provides new insights into the complex slab-weak layer interaction, it also showed some deficiencies of the modelled metrics of instability – calling for an improved representation of the mechanical properties.

## 1 Introduction

Dry-snow slab avalanche release is governed by failure processes within the layered snow cover. Whether a failure initiates and a resulting macroscopic crack will propagate, depends on the complex interaction between slab layers, the weak layer and to some extent the substratum, i.e. the layers below the weak layer. For example, the thickness and characteristics of the slab layers determine the magnitude of the stress at the depth of the weak layer due to a skier (Habermann et al., 2008; Monti

et al., 2016; Thumlert and Jamieson, 2014), and also how much deformation energy can be released to drive crack propagation (Gaume et al., 2014b; Heierli et al., 2008; McClung, 1979). On the other hand, a weak layer is a prerequisite for failure initiation and offers a path for crack propagation. Sigrist and Schweizer (2007) first described the interaction of slab and weak layer properties for evaluating the critical cut length. By interpreting their results in a fracture mechanical framework they concluded that the energy that has to be exceeded to fracture a weak layer depends on the material properties of the weak layer, whereas the energy that is available for crack propagation mainly depends on the material properties of the overlaying slab, and may also depend on the collapse height of the weak layer. Given the two most relevant processes in dry-snow slab avalanche release, failure initiation and crack propagation (e.g., Schweizer et al., 2003a), van Herwijnen and Jamieson (2007) suggested a conceptual model on the effect of the slab properties, in particular slab depth, on these failure processes. With increasing slab depth conditions for failure initiation become less favourable whereas conditions for crack propagation become more favourable.

Temporal changes in snow instability hence stem from changes in slab and weak layer properties – separately or in combination. For example, during a snowfall the probability of failure initiation in the weak layer increases due to the additional load. However, the additional load will also promote strengthening of the weak layer (e.g., Zeidler and Jamieson, 2006a) – though the strength increase may lag behind loading during a snowfall. Changes of weak layer strength have been studied in detail (e.g., Chalmers and Jamieson, 2001; Gauthier et al., 2010; Jamieson and Johnston, 1999; Zeidler and Jamieson, 2006a, b) and more frequently than changes of slab properties. However, to the best of our knowledge, there are hardly any studies that investigated how temporal changes affect the complex interplay between slab and weak layer properties.

Repeated measurements of weak layer shear strength revealed how various types of weak layers gain strength over time (Jamieson et al., 2007). Overall, an increase in strength was almost always observed in all studies while at the same time the load usually increased. Jamieson and Schweizer (2000) reported the shear strength over time for 19 buried surface hoar layers. Typical weak layer strength gain was on the order of 100 Pa d$^{-1}$ during the initial weeks after burial (Schweizer et al., 1998). Occasionally measured decreasing strength with time was attributed to spatial variability within the study site or errors associated with measurement technique (Jamieson and Johnston, 1999).

Many of the above mentioned studies monitoring strength changes of weak layers focussed on relating stability trends with observed local or regional avalanche activity (e.g., Jamieson et al., 2007). However, observed avalanche activity was often not related to the stability index, i.e. the ratio of strength to stress (e.g., Föhn, 1987), calculated from the study plot measurements. While the shear strength of the weak layer is important for failure initiation, dry-snow slab avalanches release due to crack propagation which requires the release of deformation energy stored in the slab layers. This conceptual mismatch has long been recognized. For example, Schweizer et al. (1998) pointed out that since the shear frame measurements will primarily provide information on the strength and strength changes of weak layers, the 'effective reactivity (propagation potential)' depending on the slab characteristics should be assessed by supplementary tests. Nevertheless, temporal changes of slab characteristics were rarely monitored, apart of course, from the load.

The temporal evolution of shear strength has been modelled for persistent and non-persistent weak layers (e.g., Chalmers, 2001; Conway and Wilbour, 1999; Hayes et al., 2005; Lehning et al., 2004; Zeidler and Jamieson, 2006a, b) so that the evolution of the stability index can be monitored or even forecasted (Giraud, 1993; Vernay et al., 2015). Conway and Wilbour (1999) suggested a model for natural avalanches during storms by comparing the load to the strength by assuming that failure occurs at the base of the new snow layers; their strength solely depends on density which increases with increasing overburden stress. Föhn and Hächler (1978) exclusively focused on slab properties as they studied how the slab layers settle during major snow storms. They proposed to follow the settlement (coefficient) over time to assess the probability of large natural dry-snow avalanches.

With the development of the propagation saw test (PST; Gauthier and Jamieson, 2006), now a well-established snow instability test exists providing a quantitative test result, the critical cut length. Furthermore, the PST allows determining the relevant slab and weak layer properties (Reuter et al., 2015; Schweizer et al., 2011; Sigrist and Schweizer, 2007; van Herwijnen and Heierli, 2010; van Herwijnen et al., 2016). Recently, Birkeland et al. (2014) repeatedly performed propagation saw tests on a layer of buried surface hoar; they focussed on conditions for fracture arrest.

A number of studies have focussed on the temporal evolution of spatial patterns on small uniform slopes – inter alia testing the hypothesis that variability should increase in the absence of major external forcing such as a snowfall (Birkeland and Landry, 2002; Birkeland et al., 2004; Logan et al., 2007). Hendrikx et al. (2009) used the Extended Column Test (Simenhois and Birkeland, 2009) to investigate spatial variations of the propagation potential. They assessed the spatial variability of two sites each on two days and found increased spatial clustering on the second sampling day.

For clarification, we shortly define the meaning of the following two terms: snow instability and crack propagation propensity. As Reuter et al. (2015) have pointed out, both, high failure initiation, and, high crack propagation propensity are required to describe unstable snowpack conditions. More recently, Reuter et al. (2016b) suggested to complement the failure initiation and crack propagation criteria with a tensile criterion related to dynamic crack propagation. Hence, snow instability cannot be assessed with a single stability criterion as suggested in the past (e.g., Föhn, 1987) but only by a combination of indices related to the essential processes in dry-snow slab avalanche release. With regard to crack propagation propensity, we refer to this term to describe (1) in general, whether snowpack characteristics favour self-sustained crack propagation possibly resulting in a snow slab avalanche, and (2) when interpreting propagation saw tests, whether the critical crack length is less than about one third of the column length and the crack propagates to the end of the column.

The aim of the present study is to repeatedly measure the slab and weak layer properties in a study plot to monitor their temporal evolution and to investigate their interaction in view of assessing snow instability. During the winter 2014-2015 we followed a layer of faceted crystals that was responsible for wide-spread avalanche activity in the region of Davos (Eastern Swiss Alps) over the course of two months. We performed propagation saw tests, which we analysed based on the video recordings of the tests and compared the results to concurrently performed measurements of penetration resistance using the snow micro-penetrometer (SMP) (Schneebeli and Johnson, 1998). As we performed our measurements in a level study plot equipped with an automatic weather station, we also simulated the evolution of snow stratigraphy with the

numerical snow cover model SNOWPACK. Finally, we compared our observations to metrics of instability derived from either the SMP signals or simulated with SNOWPACK. The acquired dataset provides a comprehensive time series of quantitative measures of snow instability; it allows insight into the complex interplay between slab and weak layer properties that jointly govern snow instability.

## 2 Methods

We followed the evolution of a weak layer of faceted crystals in the level study plot surrounding the automatic weather station WAN7 (2442 m a.s.l.) located in the Steintälli field site above Davos, eastern Swiss Alps (46.808° N, 9.788° E). Measurements were performed on eight days between 6 January and 3 March 2015, typically once a week during the two month study period (Table 1).

### 2.1 Weak layer formation

On 1 December 2014 the manually observed snow profile at the study plot Weissfluhjoch (2540 m a.s.l.) (located 3 km to the northeast of WAN7) showed a melt-freeze crust with 1 cm recently fallen snow on top. On 2-3 December 2014, a minor storm accumulated an additional 12 cm of new snow. During the following two weeks, the snow above the crust settled and transformed into a layer of faceted crystals due to near-surface faceting (Birkeland, 1998); this layer of faceted crystals was buried by a snowfall on 16 December 2014. The 2-3 cm thick melt-freeze crust was consistently found throughout the winter below the layer of faceted crystals that formed the 5-8 cm thick weak layer. As of mid January 2015 the layer was classified as rounded facets (FCxr) with a grain size of 1-1.5 mm. Above the weak layer was another layer of faceted crystals overlain by well consolidated slab layers that had formed in late December 2014 and early January 2015 (Figure 1). The weak layer was likely responsible for wide-spread avalanche activity in the region of Davos on 30-31 December 2014 when many natural dry-snow slab avalanches were observed. Since no snow profiles at fracture lines were recorded, we do not know the depth and type of failure layer. However, the weak layer we monitored consistently failed in snow instability tests in early January 2015, and there were no other prominent weak layers within the snow cover. This observation suggests that it was also the primary failure layer of the late-December avalanches.

### 2.2 Field measurements

On each of the eight sampling days we observed a manual snow profile, including layer density, according to Fierz et al. (2009); all profiles can be found in the supplementary material. The detailed density profile is required for the analysis by particle tracking velocimetry (PTV) of the PSTs (see below). The manual snow profile served as reference for most other measurements and was completed with the two snow instability tests that can easily be performed in flat terrain: the compression test (CT) (Jamieson, 1999) and the extended column test (ECT) (Simenhois and Birkeland, 2009).

On each sampling day, except for 19 February 2015, we also conducted at least three propagation saw tests in the immediate vicinity of the snow profile. The tests were performed according to Greene et al. (2010), albeit with longer columns. Initially, column length was around 1.5 m, and it increased towards 2 m at the end of the study period as the weak layer became more deeply buried (slab thickness increased from 59 to 148 cm) (Bair et al., 2014; Gaume et al., 2015). Using a 2-mm thick snow saw, we cut the layer of faceted crystals at its upper interface, where CT and ECT results indicated that the failure occurred. Black markers (2.5 cm in diameter) were inserted into the snowpack; we filmed all tests with a video camera for subsequent analysis by particle tracking velocimetry (van Herwijnen et al., 2010). The video recording also allowed us to accurately determine the critical value of the crack length $r_c^{OBS}$, when the crack started to propagate. Furthermore, we could also assess whether the tests were properly performed, e.g., whether the saw cut was within the weak layer. In some cases, we had to discard a test result since the cut was not performed consistently close to the layer interface; this resulted in only two values of the critical cut length on 21 January and 3 February 2015.

In the PST experiments, cracks did not always propagate to the end of the column, but arrested with a slab fracture (Table 1). These propagation results are termed END and SF, respectively (e.g., Greene et al., 2010). In the following, while reporting the critical cut length when crack propagation initiates, we do not differentiate between these two propagation results since the critical crack length is independent of the subsequent phase of dynamic crack propagation. Whether or not a running crack will arrest may depend on the tensile strength of the slab (Gaume et al., 2015; Schweizer et al., 2014). However, the onset of crack propagation entirely depends on the balance between the energy available for fracture, i.e. the mechanical energy released due an incremental advance of the crack, and the specific fracture energy, i.e. the energy required for crack growth, or in other words the resistance to crack propagation.

Concurrently, we performed several SMP measurements (SMP version 2), at least three at the location of the manual snow profile (see supplementary material), and at least one at each of the PST locations; thus in total at least 6 measurements per sampling day. The SMP measurements at the PST locations were conducted before isolating the columns, close to the end of the column where the saw cut was initiated. This procedure allowed comparing the SMP-derived properties with those from the PTV analysis.

## 2.3 PTV analysis

Using a particle tracking velocimetry algorithm (PTV; Crocker and Grier, 1996) the displacement field of the slab prior to crack propagation was calculated from the video image; it shows bending of the slab due to the saw cut (of length $r$) during the PST. Based on the displacement field of the slab, the effective elastic modulus of the slab $E^{*PTV}$ and the specific fracture energy of the weak layer $w_f^{PTV}$ were determined (van Herwijnen and Heierli, 2010; van Herwijnen et al., 2016). The approach is based on the work by Heierli et al. (2008) who suggested for the geometry of the PST (of unit width) an expression for the total energy of the system $V(r)$ as a function of crack length $r$ which consists of the fracture energy $V_f(r)$ and the mechanical energy $V_m(r)$:

$$V(r) = V_f(r) + V_m(r) = w_f r + V_m(r). \tag{1}$$

The mechanical energy includes two terms, a fracture mechanical and a bending term:

$$V_m(r) = -\frac{\pi\gamma r^2}{4E'}(\tau^2 + \sigma^2) - \frac{r^3}{6E'D}[\lambda_{\tau\tau}\tau^2 + \lambda_{\sigma\tau}\tau\sigma + \lambda_{\sigma\sigma}\sigma^2] \qquad (2)$$

where $w_f$ is the specific fracture energy, $D$ is the slab thickness, $\gamma$ is a constant of about one, depending on Dundur's elastic mismatch parameter, $E' = E/(1 - v^2)$ is the plane strain elastic modulus of the slab, and $v$ the Poisson's ratio (assumed $v = 0.2$). The load of the slab on the weak layer consists of the shear stress $\tau = -\rho g D \sin\theta$ and the normal stress $\sigma = -\rho g D \cos\theta$. where $\rho$ is the slab density and $\theta$ is the slope angle. Furthermore,

$$\lambda_{\tau\tau} = 1 + \frac{9}{4}\eta\left(\frac{r}{D}\right)^{-1} + \frac{9}{4}\eta^2\left(\frac{r}{D}\right)^{-2}, \qquad (3)$$

$$\lambda_{\tau\sigma} = \frac{9}{2}\eta + \frac{9}{2}\eta^2\left(\frac{r}{D}\right)^{-1}, \qquad (4)$$

$$\lambda_{\sigma\sigma} = 3\eta^2 + \frac{9}{4}\eta\frac{r}{D} + \frac{9}{5}\left(\frac{r}{D}\right)^2. \qquad (5)$$

with $\eta = \sqrt{4(1 + v)/5}$ .

However, by comparing estimates obtained with the analytical expression for the mechanical energy provided by Heierli et al. (2008) with finite element (FE) simulations, van Herwijnen et al. (2016) recently showed that Eq. 2 underestimates the mechanical energy for realistic values of the ratio of crack length to slab thickness $r/D$. Therefore, they introduced a correction factor that accounts for the sensitivity to $r/D$ and the slope angle $\theta$ to obtain an adjusted mechanical energy $V_m^*(r)$.

According to theorem of Clapeyron, for a linear-elastic system, the mechanical energy can be estimated from the gravitational potential energy. The gravitational potential energy was computed from the measured displacement field, assuming a layered slab and using the manually observed density profile, to determine the mechanical energy as a function of crack length. This measured mechanical energy was then fitted to the adjusted mechanical energy $V_m^*(r)$ to obtain the effective elastic modulus of the slab $E^{*PTV}$, the fit parameter. It is defined as the modulus of a uniform slab of equal mean density yielding the same displacement as the real slab. To determine the specific fracture energy $w_f^{PTV}$, the analytical expression for the adjusted mechanical energy with the best fit modulus is differentiated with regard to the crack length at the critical cut length $r_c^{OBS}$. In other words, the slope of $V_m^*(r)$ at the critical crack length corresponds to the specific fracture energy of the weak layer. For a more detailed description on deriving the effective elastic modulus and the specific fracture energy, the reader is referred to van Herwijnen et al. (2016).

## 2.4 SMP signal processing

We used the penetration resistance data acquired with the SMP to obtain detailed data on the layering of the snow cover to derive mechanical properties following the approach described in Reuter et al. (2015).

Based on the manually observed snow profile, layers were manually defined from the corresponding sections of the SMP signal, i.e. several slab layers, a weak layer and a basal layer. The shot-noise model by Löwe and van Herwijnen (2012)

was then applied to determine snow micro-structural parameters, namely the rupture force $f$, the deflection at rupture $\delta$ and the structural element size $L$. These three parameters were calculated over a moving window of 2.5 mm with 50% overlap and averaged over each layer. Furthermore, for each layer the snow density was derived according to Proksch et al. (2015), allowing to calculate the load on the weak layer. The micro-mechanical modulus for the slab layers and the strength of the

weak layer $\sigma_{\mathrm{WL}}$ were calculated according to Johnson and Schneebeli (1999). The weak layer specific fracture energy $w_{\mathrm{f}}^{\mathrm{SMP}}$ was calculated as the minimum of the integrated penetration resistance across each moving window within the weak layer (Reuter et al., 2013). Finally, the penetration depth was estimated by integrating the penetration resistance $F$ from the snow surface to the depth of penetration where a threshold value of the absorbed energy was reached (Schweizer and Reuter, 2015).

10        The effective elastic modulus of the slab $E^{*\mathrm{SMP}}$ was determined by performing FE simulations of the experimental setup taking into account the slab layering (for details see Reuter et al., 2015). The FE model consisted of all the slab layers, the weak layer and a basal layer, each with density, micro-mechanical modulus and thickness values derived from the SMP measurement. The mechanical strain energy $V_{\mathrm{m}}^{\mathrm{FEM}}(r)$ was then calculated for a stratified slab bending over a crack of increasing length $r$. In order to recover an effective elastic modulus $E^{*\mathrm{SMP}}$, the analytical expression for the adjusted

mechanical energy $V_{m}^{*}(r)$ was fitted to the modelled values of mechanical energy $V_{\mathrm{m}}^{\mathrm{FEM}}(r)$. Hence, we followed the same approach as for the PTV analysis, and we also used the newly developed correction factor to obtain the adjusted mechanical energy $V_{m}^{*}(r)$.

        The SMP-derived weak layer specific fracture energy and the effective elastic modulus were scaled by a linear factor of 2.8 and 2.5, respectively, to approximately match the corresponding values derived from the PTV analysis. Scaling

of the specific fracture energy and the modulus was performed as there is no calibration yet of the microstructural mechanical properties that can be derived from the SMP signal. Whereas the microstructural properties derived from the SMP are physically based, they cannot be expected to directly represent the corresponding macroscopic properties (Reuter et al., 2016a).

        In addition, an alternative effective elastic modulus $E_{\rho}^{*\mathrm{SMP}}$ was derived, following the same approach with the same

FE model as described above. However, rather than using the micro-mechanical modulus, for each layer the modulus was determined using the SMP-derived density and applying the parametrization provided by Scapozza (2004):

$$E = 1.873 \times 10^{5} e^{0.0149\rho}.$$

        Based on the mechanical parameters estimated from the SMP measurements, two metrics of point instability were derived, as suggested by Reuter et al. (2015).

30        The first metric is the failure initiation criterion $S$, a strength-over-stress criterion describing the propensity of the weak layer to fail in case of skier loading:

$$S = \frac{\sigma_{WL}}{\Delta\tau_{\mathrm{FEM}}} \tag{6}$$

where $\sigma_{WL}$ is the SMP-derived micro-mechanical strength of the weak layer and $\Delta\tau_{FEM}$ the maximum additional shear stress at the depth of the weak layer due to skier loading. The maximum shear stress at the depth of the weak layer was modeled with the 2D linear elastic FE model originally presented by Habermann et al. (2008) to calculate the shear stress $\Delta\tau_{FEM}$ below a layered slab due to the weight of a skier.

The second metric is the crack propagation criterion $r_c^{SMP}$, an SMP-derived critical cut length. It is derived by numerically solving Eq. 7, which is obtained by finding the extremum of the total energy of the cracked system $V(r)$ (Eq. 1) with respect to $r$ and ensuring that it is a maximum: $\frac{d}{dr}V(r) = w_f + \frac{d}{dr}V_m(r) = 0$, which yields (Schweizer et al., 2011):

$$w_f(E', r_c) = \frac{D}{2E'}\left[w_0 + w_1\frac{r_c}{D} + w_2\left(\frac{r_c}{D}\right)^2 + w_3\left(\frac{r_c}{D}\right)^3 + w_4\left(\frac{r_c}{D}\right)^4\right], \tag{7}$$

with

$$w_0 = \frac{3\eta^2}{4}\tau^2,$$

$$w_1 = \left(\pi\gamma + \frac{3\eta}{2}\right)\tau^2 + 3\eta^2\tau\sigma + \pi\gamma\sigma^2,$$

$$w_2 = \tau^2 + \frac{9\eta}{2}\tau\sigma + 3\eta^2\sigma^2,$$

$$w_3 = 3\eta\sigma^2,$$

$$w_4 = 3\sigma^2.$$

By inserting the effective elastic modulus of the slab $E_\rho^{*SMP}$ and weak layer specific fracture energy $w_f^{SMP}$ into Eq. 7 the crack propagation criterion $r_c^{SMP}$ was obtained. Hence this modelled critical cut length is independent of the critical cut length $r_c^{OBS}$ measured in the field. For a more detailed description on how to obtain the above mentioned mechanical properties as well as the SMP-derived metrics of point instability, the reader is referred to Reuter et al. (2015).

**2.5 Snow cover modelling**

We compared results from field measurements to output of the numerical snow cover model SNOWPACK (e.g., Lehning et al., 2004) driven by meteorological input from the automatic weather station WAN7. This weather station is located in the study plot where we performed the field measurements. Meteorological input contained air temperature and relative humidity (Rotronic MP100H HygroClip, ventilated), wind speed and direction (YOUNG wind monitor), incoming short and

long wave radiation (Campbell CNR1), and snow height (Campbell SR50). Data gaps shorter than one day were filled by linear interpolation. Gaps longer than one day were filled with data from the nearby AWS Weissfluhjoch (2540 m a.s.l.; 3 km to the northeast). Variables were filtered by introducing reasonable lower and upper limits, e.g. 5 and 100 % for relative humidity. The modelling time step was 15 min after resampling the data from the AWS with a sampling rate of 10 min. The model was initiated on 1 October 2014, when no snow was present at the AWS and ran until the end of May.

Neumann boundary conditions for estimating the snow surface temperature and atmospheric stability corrections for estimating turbulent exchange were the preferred adjustments concerning the energy balance model (Stössel et al., 2010).

From the model output, we evaluated the skier stability index SK38 introduced by Jamieson and Johnston (1998). It is defined as the ratio of shear strength to shear stress: $SK38 = \frac{\tau_s}{\tau + \Delta\tau}$ with $\tau$ the shear stress due to the weight of the overlaying slab, $\Delta\tau = 155/h$ the additional shear stress due to a skier with $h$ the slab depth (Föhn, 1987; Monti et al., 2016), and $\tau_s$ the shear strength as parameterized with density and grain type according to Jamieson and Johnston (2001).

In addition, we estimated the critical cut length $r_c^{\text{SNP}}$ from the snow stratigraphy provided by SNOWPACK. We used SNOWPACK model output to derive all the required mechanical properties of the snow layers. The critical cut length was then estimated based on the relation given by Gaume et al. (2014a, Eq. 5) (see Gaume et al., 2016 for a detailed derivation). Based on discrete element modelling they suggested the critical cut length, in the flat (for slope angle $\theta = 0°$), to essentially depend on the plain strain elastic modulus of the slab $E'$, slab load $\sigma$, and weak layer shear strength $\tau_s$ :

$$r_c^{\text{SNP}} = \Lambda\sqrt{\frac{2\,\tau_s}{\sigma}} \qquad\qquad (8)$$

with $\Lambda = (E'\,D\,D_{\text{WL}}/G_{\text{WL}})^{1/2}$, $D$ the slab thickness, $D_{\text{WL}}$ the weak layer thickness and $G_{\text{WL}}$ its shear modulus. All the required parameters are provided by SNOWPACK, except for the elastic moduli $E'$ and $G_{\text{WL}}$. For the shear modulus of the weak layer we assumed a constant value of 0.5 MPa, based on laboratory experiments (Camponovo and Schweizer, 2001; Reiweger et al., 2010). For the elastic modulus of the slab, we followed the same FE approach as described above for the SMP analysis to derive an effective elastic modulus taking into account slab layering rather than using a slab modulus based on the average slab density. Hence, for each layer of the modelled snow stratigraphy an elastic modulus was calculated from density based on the relation provided by Scapozza (2004). With these properties (modulus, layer density and thickness) a FE simulation was performed to determine the effective elastic modulus $E^{*\text{SNP}}$.

## 2.6 Avalanche activity

Study plot measurements are commonly used in operational forecasting to make assessments about the avalanche danger in the surrounding terrain (e.g., Gauthier et al., 2010). We therefore compared the results obtained from the field measurements to the observed avalanche activity in the region of Davos. These observations include the number, type and size of avalanches recorded by personnel from the local ski areas, the avalanche warning service, SLF staff members, and others. We then calculated the avalanche activity index per day as described by Schweizer et al. (2003b). The index is a weighted sum of the number of observed avalanches per day including natural as well as artificially triggered avalanches; the weights depend on the avalanche size and are 0.01, 0.1, 1 and 10 for Canadian size classes 1 to 4, respectively (McClung and Schaerer, 2006).

**2.7 Case studies**

To explore the complex interaction between slab and weak layer properties on the critical cut length in a PST, we considered a few cases with exemplary temporal evolutions of slab load, slab elastic modulus and weak layer specific fracture energy. To this end we numerically solved Eq. 7 to obtain the corresponding critical crack length, and hence its evolution over time. These examples are not meant as a sensitivity study where one parameter is varied and the other held constant, but as an illustration of the interaction when most or all parameters change.

**3  Results**

**3.1 Propagation saw test results**

On 6 January 2015, when we did the first propagation saw tests, cracks did not always fully propagate to the end of the column, but slab fractures were observed in three tests (Table 1, Figure 2a). Eight days later, on 14 January 2015, when the PSTs were performed on a slightly more shallow part of the study plot surrounding the automatic weather station WAN7, all five tests resulted in slab fractures. As of 28 January 2015, 22 days after the first measurements, all cracks fully propagated to the end of the column. After 56 days, on 3 March 2015, in all three tests cracks still fully propagated while the critical crack length had increased to about 50 cm. By then, the weak layer of faceted crystals was buried below a slab of about 150 cm in thickness with an average density of about 270 kg m$^{-3}$, resulting in a load of almost 4 kPa.

During our measurement series, the critical cut length $r_c^{OBS}$ was initially about 23-30 cm, and slightly increased up to 21 January 2015. Consistently shorter cut lengths were observed on the following measurement day, on 28 January 2015, the date when all cracks fully propagated for the first time. On the following sampling day, 3 February 2015, one test again yielded a short cut length, 17 cm, the shortest value recorded during our sampling period. Subsequently, the cut lengths increased with time.

**3.2 Load**

On 6 January 2015, the weak layer was buried below a slab of 59 cm (total snow depth HS = 115 cm) and the initial load was about 1.4 kPa (Table 1, Figure 2b). The load did not change much during the following week but then continuously increased due to snowfalls to almost 3 kPa by early February 2015. After a fair weather period in February with no new snow for more than two weeks, the snow depth increased again, and on the last sampling day (3 March 2015) the load was about 3.9 kPa.

The density derived from the SMP signal agreed well with the manually measured density (not shown) and accordingly the increase in load above the weak layer was well represented by the SMP measurements (Figure 2b). For the numerical snow cover model SNOWPACK, values of load as calculated from average density and slab thickness were often

slightly lower than those measured in the field. The underestimation is mainly due to the fact that the modelled snow depth was about 25 cm lower than measured at the location of the manual snow profiles.

**3.3 Effective elastic modulus of the slab**

The effective elastic modulus of the slab was derived from the bending of the slab during the propagation saw test via the PTV analysis $E^{*\mathrm{PTV}}$ (Figure 3a) as well as from the SMP signal analysis using the FE model with either the micro-mechanical modulus or a modulus derived from SMP density, yielding $E^{*\mathrm{SMP}}$ and $E_\rho^{*\mathrm{SMP}}$, respectively (Figure 3b). The effective elastic modulus of the slab obtained from the PTV analysis showed an overall increase from about 2.5 to 10 MPa during the two-month sampling period. However, the increase was not steady; for example, on 13 February 2015 relatively low values between 2.4 and 6.1 MPa were obtained.

The SMP-derived effective elastic modulus $E^{*\mathrm{SMP}}$ initially did not change much with median values of approximately 3 MPa during the first 16 days. It then increased until the end of January to about 5 MPa. However, subsequently, it decreased and on the last measuring day low values of only about 1.8 MPa were derived.

The alternative SMP-derived modulus $E_\rho^{*\mathrm{SMP}}$ also increased from initially 8 MPa to about 22 MPa in early February. Thereafter $E_\rho^{*\mathrm{SMP}}$ did not change much anymore with values between 19 and 26 MPa (median values per day).

**3.4 Weak layer specific fracture energy**

The PTV analysis suggests that the weak layer specific fracture energy $w_\mathrm{f}^{\mathrm{PTV}}$ overall increased with time from about 0.6 J m$^{-2}$ to about 2.2 J m$^{-2}$, mostly after the end of January 2015 (Figure 4a). The SMP analysis revealed a similar tendency of increasing weak layer specific fracture energy with time. Overall, $w_\mathrm{f}^{\mathrm{SMP}}$ increased from about 0.5 J m$^{-2}$ to 1.4 J m$^{-2}$ and most of the increase occurred towards mid February (Figure 4b).

**3.5 SMP-derived metrics of instability**

The failure initiation criterion $S$ derived from SMP resistance data was initially about 300, indicating a transitional value for failure initiation propensity given the threshold reported by Reuter et al. (2015). They observed that a value of the initiation criterion of about 230 divided between the cases with and without concurrently observed signs of instability. The index then increased to about 600 towards the end of January and further to about 1100 by the end of the sampling period (Figure 5a).

The SMP-derived critical cut length $r_\mathrm{c}^{\mathrm{SMP}}$ overall increased from about 40 cm to 70 cm (Figure 5b). Again the increase was not steady with a similar tendency with time as shown for the weak layer specific fracture energy $w_\mathrm{f}^{\mathrm{SMP}}$. At the three sampling days between end of January and mid February similar values of $r_\mathrm{c}^{\mathrm{SMP}}$ were derived, namely about 55 cm.

### 3.6 SNOWPACK derived metrics of instability

The skier stability index SK38 (Figure 5c) for the investigated weak layer showed values between 1.1 and 1.35 in early January 2015; these values are in the range of transitional stability (1 to 1.5) according to Jamieson and Johnston (1998). With the snowfalls at the end of January and early February (Figure 1) the index then decreased towards 1.0 indicating increased triggering probability. Subsequently, during much of February, when there was no snowfall, the skier stability index SK38 gradually increased towards 1.3 and decreased again to overall the lowest values of about 1 towards the end of the observation period.

The modelled critical cut length $r_c^{SNP}$ (Figure 5d), on the other hand, steadily increased from about 20 cm in early January to 60 cm in early March 2015.

### 3.7 Avalanche activity

The highest avalanche activity in the region of Davos was observed around the end of the year 2014 (Figure 6), before our first measurements were performed. In January 2015, avalanches were occasionally observed, in particular on 18 January, a sunny Sunday after a snowfall. Towards late January and early February, there was a period of increased avalanche activity. The last peak was on 3 March 2015, again after a major snowfall, but during this period avalanches did no longer run on the weak layer of facets we monitored; the observed fracture depths were much lower than the burial depth of the weak layer we investigated. Overall, avalanche activity clearly decreased since early January 2015. However, after each significant snowfall avalanches were again observed (e.g. on 18, 28 and 31 January, Figure 6), i.e. the weak layer of facets was still reactive, a true persistent weak layer.

The overall decrease in avalanche activity until the end of February is in line with the observed results of the CTs and ECTs we performed concurrently on each sampling day. The scores increased from values just below 20 taps to values of around 30 taps (red asterisks in Figure 6). On the first measuring day, no crack propagation was observed with the extended column test (Table 1). Subsequently, the fracture type in ECTs was always P (propagation across the entire column), except on 28 January when one ECT did not fracture (X). The increased avalanche activity towards the end of January and early February coincides with the lowest values of observed critical cut length.

### 3.8 Case studies

The observed temporal evolution of the critical cut length in our PSTs showed an unsteady increase with the lowest values in the middle of the measuring period and an increase thereafter. To explore whether this type of pattern is possible at all, we numerically solved Eq. 7 to find out how the critical cut length changes with time for various scenarios of the temporal evolution of the specific fracture energy of the weak layer $w_f$, the load $\sigma$ and the modulus of the slab $E$.

In the first simplified scenario, all input parameters were assumed to increase, corresponding to a situation when the slab thickens and becomes stiffer, while at the same time the weak layer toughens due to the increased load (Figure 7a). In

this scenario, the critical cut length did almost not change, showing a very slight increase (Figure 7b). The combination of increasing load and increasing slab modulus provided a bit more energy to drive the crack, but just as much about to compensate the increased specific fracture energy of the weak layer. This indicates that in this scenario the load had a larger effect than the modulus, since an increasing modulus reduces the amount of strain energy available due to less deformation.

5          In the second simplified scenario, the specific fracture energy of the weak layer increased as the load increased, while the slab modulus remained constant (Figure 7c). This scenario resulted in a decreasing critical cut length (Figure 7d). Due to the increasing load, more energy was available to drive the crack, which outweighed the additional energy required to advance the crack due to the weak layer toughening.

          In the third simplified scenario, the load remained constant, while the slab modulus and the weak layer specific
fracture energy increased (Figure 7e); this scenario corresponds to a fair weather period where the slab stiffens due to settlement and the weak layer toughens due to ongoing sintering with time. In this scenario, the critical cut length substantially increased (Figures 7f). Due to the higher slab modulus less energy was released to drive crack propagation while at the same time the weak layer became tougher.

          Finally, in our last scenario, we attempted to roughly mimic the temporal evolution of the critical cut length as
observed in our PSTs. We assumed the load to increase approximately as observed, the modulus to triple and the specific fracture energy of the weak layer to increase as well, but not continuously (Figure 7g). With these assumptions, the critical cut length first increased, had a local minimum approximately in the middle of the considered time period and finally increased again (Figures 7h).

## 4  Discussion

We repeatedly performed propagation saw tests in a level study plot to follow the temporal evolution of a weak layer of faceted crystals and of the overlaying slab layers by combining state-of-the-art measurement techniques and numerical snow cover simulations. Specifically, we used particle tracking velocimetry, the snow micro-penetrometer and the snow cover model SNOWPACK to estimate snow mechanical properties and derive snow instability criteria.

### 4.1 Observed critical cut length (PST)

While performing the propagation saw tests in the study plot, we initially observed a mixture of END and SF test results; mixed test results are typically not found in critical conditions. Only when the load had reached 2 kPa, all cracks fully propagated towards the end of the column. This observation suggests that the tensile strength of the slab was initially not large enough to allow crack propagation (Gaume et al., 2015), in particular at the more shallow locations. Other changes in slab properties that might have favoured slab fractures, e.g. faceting, were not observed.

30          The initial absence of full crack propagation in our field tests contrasted with the high avalanche activity observed during that time. The contrasting observations may be due to the fact that we made our measurements in a wind-sheltered

study plot that may not be very representative of wind-affected starting zones in the area. In typical lee-slope starting zones, the tensile strength of the slab might have been larger; moreover, the tensile stress might be lower on slopes since less bending is expected prior to natural avalanche release than observed in PSTs in flat terrain (McClung and Borstad, 2012). In general, slab layers are more variable in terms of penetration resistance than weak layers reflecting the dynamic conditions of

snowfall and wind during deposition of the slab (Schweizer et al., 2008). Since the properties of the slab layers are particularly important for crack propagation, variable crack propagation propensity has to be expected. The observed discrepancy highlights some of the limitations when extrapolating instability from flat field study sites.

As of 28 January 2015 all cracks in PSTs propagated to the very end of the column. This is in line with the results of the other stability test performed: sudden fractures (SP/SC) in CTs and full crack propagation (P) in ECTs.

Overall the observed critical cut length doubled from about 25 cm to about 50 cm by early March. However, the increase was not steady. Despite the overall increase in critical cut length, the lowest values were observed at the end of January and in early February. This pattern of the temporal evolution of the critical cut length in PSTs was rather unexpected since weak layers typically gain strength with time (e.g., Jamieson and Schweizer, 2000).

Consistently low values of the critical cut length in PSTs, between 19 and 24 cm, were observed on 28 January

2015. On the following sampling day, the lowest value (17 cm) was observed, but also a rather high value of 38 cm. In general, low values or scores in snow instability tests are more trustworthy than high values. In the case of the propagation saw test, any measurement and observation errors increase the cut length. In our case, we were cutting at the top of a weak layer of faceted crystals, and the layer above was not much harder and as well consisted of faceted crystals. Hence, it was relatively easy to move the saw out of the weak layer. We were able to identify such deviations on the video recordings, but

only on the side facing the camera. The high values may therefore be due to imperfect sawing and show the difficulty in obtaining consistent PST results in weak layers which are not very well defined.

The median range, i.e. the difference between the highest and the lowest value, of the PST results on a given sampling day was 5.9 cm, so that the resulting uncertainty in the mean is about 2-3 cm. Reuter and Schweizer (2012) reported a standard deviation of the critical cut length on days with surface warming of about 5 cm. Similar variations for

PST results on a single day at a single location have been reported by Gauthier and Jamieson (2008).

Therefore, we have no reason to dismiss these low values or attribute them to measurement errors. However, they may be related to spatial variations in weak layer and slab properties within the study plot. In fact, on 14 January 2015 we performed the PSTs at a location were the snow depth was below average compared to the rest of the study plot. On all other days snow stratigraphy was very similar, exemplified by mostly similar SMP profiles (not shown). The observation that

snow stratigraphy in study plots is often spatially not particularly variable has, for instance, been shown by Pielmeier and Schneebeli (2001). Previous snow instability studies performed in level study plots suggested that measurements in general are reliable and that the effect of spatial variations is relatively small. Jamieson (1995) reported a mean coefficient of variation of shear strength of about 15% for sets of measurements in study plots. Correspondingly, variations in stability indices derived from study plot measurements of load and shear strength, two measurements that have comparable errors as

the PST, were found to be indicative of avalanche activity (e.g., Jamieson et al., 2007). Though the PST results may be influenced by some small scale spatial variability of the snowpack in the study plot, we deem it unlikely that the observed minimum values are entirely the result of spatial variability, but indicate in fact a period of high propagation propensity. This interpretation is supported by the increased avalanche activity observed in late January and early February when many skier-triggered avalanches were observed (Figure 6).

## 4.2 Load, modulus and specific fracture energy

The temporal evolution of the parameters influencing the critical cut length, namely the load, the effective elastic slab modulus and the weak layer specific fracture energy, all exhibited similar, mostly increasing trends. The load obviously increased (Fig. 2b) and the results derived from the manual density measurements and the SMP profiles agreed well – in accordance with recent comparison studies (Proksch et al., 2015; Proksch et al., 2016).

The PTV-derived effective elastic modulus also increased, but the increase was not steady with some low values on 13 February 2015 (Figure 3a). These low values were also observed with the SMP (Figure 3b). The SMP-derived effective elastic modulus $E^{*\mathrm{SMP}}$, however, showed very low values on the very last sampling days – resulting in an overall decrease. When estimating the effective elastic modulus from the SMP-derived density, the agreement in temporal evolution was better. Indeed, overall $E_\rho^{*\mathrm{SMP}}$ also increased. The observation that the SMP-derived modulus occasionally decreases compared to the previous measurement day, for example, on 3 February and 3 March 2015 can be explained by the fact that on each of these days a snowfall period ended. With the addition of new snow on the top of the existing slab, the effective elastic modulus, considering the FE model, in general decreases, unless the old slab layers below substantially settle so that the penetration resistance increases. In other words, the additional load due to the new snow leads to a thicker slab of lower average density, but also resulting in a lower effective elastic modulus.

In general, the weak layer specific fracture energy is expected to increase with time as sintering will increase bonding between crystals – unless the weak layer is subject to a large temperature gradient (e.g., van Herwijnen and Miller, 2013). Indeed, the PTV-derived weak layer specific fracture energy $w_\mathrm{f}^{\mathrm{PTV}}$ and the microstructural related specific fracture energy derived from the SMP signal $w_\mathrm{f}^{\mathrm{SMP}}$ exhibited a similar overall increasing trend (Figure 4). However, independent of the evaluation method, the increase occurred mostly after early February. This observation suggests that the weak layer toughening mainly occurred during the fair weather period without additional loading in February. A constant weak layer specific fracture energy in combination with additional loading by snowfall in January would have resulted in a decreasing critical cut length, in line with field observations.

With respect to the absolute values of the specific fracture energy, these should be considered as effective values since it is clear that they are too high compared to the specific fracture energy of ice (McClung, 2015). For the PTV-derived values, the discrepancy is most likely related to the fact that the observed bending in a PST includes non-elastic parts of deformation, thereby increasing the specific fracture energy to physically unrealistic values; for an in-depth discussion of this issue see van Herwijnen et al. (2016).

Compared to the weak layer specific fracture energy, the PTV- and SMP-derived absolute values of the effective elastic modulus agreed less well. In particular, it is known that $E^{*\mathrm{SMP}}$ does not relate well to the PTV-derived modulus (Reuter et al., 2013) – even though we scaled the SMP-derived values with the corresponding PTV-derived modulus (see below). On the other hand, van Herwijnen et al. (2016) recently showed that the PTV-derived modulus fits relatively well

with results from laboratory experiments in the same range of strain rates. The SMP-derived modulus $E^{*\mathrm{SMP}}$ includes the complex interaction between the cone and surrounding snow microstructure and does obviously not reflect simple elastic deformation, but breaking, jamming and other local effects occur (LeBaron et al., 2014; van Herwijnen, 2013). Obviously, the SMP-derived modulus $E_\rho^{*\mathrm{SMP}}$, which relies on the relatively robust density derivation, agreed somewhat better with the PTV-derived modulus.

In general, when comparing the absolute values of the effective elastic modulus and the weak layer specific fracture energy, we recall that the SMP-derived values of $E^{*\mathrm{SMP}}$ and $w_\mathrm{f}^{\mathrm{SMP}}$ were scaled to the corresponding PTV values to ease comparison. The scaling was performed using a larger dataset (unpublished) of side-by-side PST and SMP measurements. It is beyond the scope of this study to provide a quantitative comparison of the two methods; it will be the subject of a publication currently in preparation (Reuter et al., 2016a). To conclude the discussion on the PTV and SMP-derived values,

below we present an error assessment – as far as this is possible.

The errors associated with the parameters derived from the PTV analysis (i.e. the measurement uncertainty) were about 20% (or about 1 MPa) for the modulus and about 18% (or about 0.14 J/m$^2$) for the weak layer specific fracture energy. These estimates are based on calculating these properties 1000 times for each experiment accounting for the uncertainty in the input parameters (uncertainty in the distance measurements in the field, density measurements, and especially the

location estimates of the dots in the PTV analysis). van Herwijnen et al. (2016) showed that the reproducibility of side-by-side measurements for the slab modulus was good (even though the values of the modulus have greater uncertainty), whereas the reproducibility for the weak layer specific fracture energy was lower. The better reproducibility for the modulus might be related to the fact that the modulus is an integrated property over all slab layers and hence may be less sensitive to spatial variations of one layer.

The errors of the SMP-derived parameters are more difficult to assess. However, Proksch et al. (2015) showed that the SMP-derived density is a reliable measure. Their finding is supported by our measurements showing good reproducibility between SMP-derived and manually measured load (Figure 2b). However, in particular the derivation of the effective elastic modulus is more prone to errors. In general, the variability of SMP-derived microstructural parameters is rather high. Thus, any value which is indirectly derived from these microstructural parameters will exhibit large variations;

in particular estimates of the deflection at rupture $\delta$ are rather variable (Löwe and van Herwijnen, 2012).

### 4.3 Metrics of instability

The SMP-derived metrics of point snow instability, the failure initiation criterion $S$ and the crack propagation criterion $r_c^{SMP}$, were recently developed and validated (Reuter et al., 2015). Here, we applied them for the first time to monitor the temporal evolution of instability. The failure initiation criterion $S$ increased with time suggesting that initiating a failure in the weak layer became increasingly difficult during the sampling period. This tendency is supported by the results of the two small column snow instability tests (Table 1, Figure 6). CT/ECT scores increased from around 20 to 30 taps. The absolute values of $S$ were rather high, in the range that was considered as rather stable by Reuter et al. (2015). Again, this is in agreement with the rather high scores of the CTs and ECTs and is related to the fact that by the end of January 2015 the weak layer was buried below a thick slab of more than 1 m. Weak layers buried deeper than 1 m are not frequently triggered by skiers (e.g., Schweizer and Jamieson, 2007).

The crack propagation criterion $r_c^{SMP}$ overall increased in agreement with the observations (Figure 2a). It showed a similar evolution with time as the SMP-derived weak layer specific fracture energy, which is used to calculate $r_c^{SMP}$. A relative decrease towards the end of January was also observed but was less prominent than for $r_c^{OBS}$. Despite substantial scatter, until early February about half of the values were between 35 and 45 cm. Only after mid February $r_c^{SMP}$ values increased to about 70 cm. Considering the threshold values indicating unstable conditions $S < 234$ and $r_c^{SMP} < 41$ cm suggested by Reuter et al. (2015), the two criteria predict unstable conditions in early January. In fact, signs of instability were observed by the field team on 6 January and 28 January 2015; on the latter day the lowest values of critical cut length were observed in PSTs. However, at that day the failure initiation criterion was already high ($S \approx 590$), since the slab thickness was >1 m.

Based on the simulated snow stratigraphy provided by SNOWPACK, we presented two corresponding metrics of instability. Overall, snow stratigraphy was well simulated, as exemplified by the comparison shown in Figure 1. Despite large differences in vertical resolution, the simulated SNOWPACK profile, the manually observed profile and the SMP profile qualitatively agreed well.

The SNOWPACK-derived skier stability index SK38 varied between about 1.0 and 1.35 and did not show any trend with time. This is in contrast to the increasing scores obtained with CTs and ECTs. Whereas the skier stability index SK38 initially increased in agreement with the observation, it subsequently mainly reflected whether there was any loading by new snow. After the end of January when the weak layer was deeply buried, the SK38 did no longer indicate skier triggering but yielded almost the same value as the natural stability index (not shown) since $\Delta\tau$, the additional shear stress due to a skier, became negligibly small. Accordingly, increasing load due to snowfall resulted in a decreasing SK38 as shown towards the end of February and in early March. In other words, the SK38 became dominated by the static shear stress. In contrast, the SMP-derived initiation criterion $S$ does not include the static shear stress, and hence showed a different behaviour with time. As Schweizer and Reuter (2015) pointed out, the dynamic load, rather than the static load due to the weight of the slab, is

essential for initiating a failure due to the well-known deformation rate dependence of snow strength (e.g., Reiweger and Schweizer, 2010).

The modelled critical cut length $r_c^{SNP}$ (Eq. 8) based on recent findings by Gaume et al. (2016) steadily increased over the two-month period. All three essential variables, namely the load, the slab modulus and the weak layer shear strength

overall increased with time. Whereas an increase of the load – all other variables remaining unchanged – would result in a decrease of the critical cut length, increasing slab modulus as well as increasing weak layer shear strength leads to increasing critical cut length. The latter two variables are directly related to snow density which in general increases with time – except for the average slab density which may temporarily decrease after a snowfall. In our case, with regard to the critical crack length $r_c^{SNP}$, the effect of increasing load was clearly over-compensated by the effects of increasing slab effective modulus

and increasing weak layer shear strength.

The discrepancy between $r_c^{SNP}$ and the observed critical cut length seems to be due to the weak layer shear strength which strongly increased with increasing density from initially 0.9 to 1.9 kPa in early March – though persistent weak layers are known to hardly settle despite increasing overburden pressure due to their anisotropic microstructure (e.g., Reiweger and Schweizer, 2010; Walters and Adams, 2014). Hence, discrepancy seems mainly due to overestimating weak layer density.

Also, the SMP analysis of the weak layer strength (not shown) suggests that the increase was not as prominent as shown by SNOWPACK where the shear strength is parameterized based on the extensive shear frame measurements by Jamieson and Johnston (2001). Whereas Gaume et al. (2016) recently showed that the modelled critical cut length $r_c^{SNP}$ of the weak layer we tested was lower than of the adjacent layers – suggesting that the modelled critical cut length can well discriminate between weak layers and other layers within a given snow stratigraphy, it seems more challenging to predict changes over

time since small changes of the contributing variables may decide whether the critical cut length increases or decreases.

## 4.4 Case studies

In general, considering the fracture mechanical approach (Anderson, 2005) reveals that in a first order approximation $r_c \sim \frac{w_f E}{\sigma^2}$ ; the critical crack length decreases with increasing load, and increases with either increasing slab elastic modulus or increasing specific fracture energy of the weak layer. To explore the complex interaction of these parameters described by

Eq. 7, we presented four case studies. They show how the temporal evolution of the load and the modulus of the slab, and the specific fracture energy of the weak layer affects the changes of the critical cut length with time. The most influential parameter seems to be the load. Even with an increasing weak layer specific fracture energy, the increasing load caused the cut length to decrease (Figures 7c,d). This consistent decrease was however not observed in the field, where only towards the end of January the critical crack length decreased and then clearly increased towards the end of the measurement period.

This suggests that the increase in slab modulus and/or specific fracture energy (over)-compensated the effect of the increasing load. The fourth case, more or less mimicking the observed changes with time of the three variables, shows that

under certain conditions the cut length may increase or decrease with time, primarily due to subtle changes of slab properties.

Alternatively, the sensitivity of the critical cut length could be explored with Eq. 8. In fact, if the temporal evolution of the shear strength followed the one assumed for the weak layer specific fracture energy, the same trends for the critical cut length were observed (not shown) confirming the findings of the case studies.

## 5  Conclusions

We monitored the temporal evolution of a weak layer of faceted crystals that was one of the critical weaknesses during the winter 2014-2015 in the region of Davos (Eastern Swiss Alps). We focused on the crack propagation propensity and performed propagation saw tests on seven sampling days during a two-month period from early January to early March 2015. Tests were completed with objective measurements, namely by resistance profiles acquired with the snow micro-penetrometer and particle tracking velocimetry based on video images of the PSTs. Our dataset represents the first comprehensive time series of quantitative measures of critical cut length and related mechanical properties of slab layers and weak layer. In addition, we compared our field observations to newly developed metrics of instability either derived from the SMP signal or from modelled snow stratigraphy.

The critical cut length, observed in the PSTs, showed an overall tendency to increase with time. However, the lowest values were observed towards the end of January/early February. These low values were not expected and seem to be the result of the complex interaction of slab and weak layer properties – rather than of measurements errors or large spatial variations within the study plot.

The relevant mechanical properties, the effective elastic modulus of the slab and the weak layer specific fracture energy, overall increased, independent of the evaluation method. Only the SMP-derived effective elastic modulus $E^{*\mathrm{SMP}}$ showed a different behaviour. However, the increase was not steady and these small changes likely affected the critical cut length as exemplified with the case studies. These findings suggest that it is not possible to assess the critical cut length, or in general crack propagation propensity, by simply monitoring a subset of these mechanical properties. One has to consider the complex interaction between the effective elastic modulus of the slab, the load due to the slab, and the weak layer specific fracture energy. Furthermore, these properties have to be determined with good accuracy since otherwise reliably modelling the critical cut length is not possible.

We then applied traditional (such as the SK38) and newly-developed metrics of snow instability describing either the failure initiation or the crack propagation propensity. These metrics were derived from the SMP signal or calculated from simulated snow stratigraphy (SNOWPACK) and partially reproduced the observed temporal patterns. Whereas the SMP-derived initiation criterion appropriately indicated that triggering probability overall decreased, the skier stability index provided by SNOWPACK did not show this tendency, but indicated the period of increased avalanche activity towards the

end of January. The predicted critical cut lengths, $r_c^{\text{SMP}}$ and $r_c^{\text{SNP}}$, overall increased with time. Whereas the SMP-derived propagation criterion $r_c^{\text{SMP}}$ partly reproduced the unsteady increase with some lower values towards the end of January, the SNOWPACK-derived critical cut length $r_c^{\text{SNP}}$ did not show any of the observed variation in critical cut length, apart from the overall increase.

Whereas the PST combined with the PTV approach seems to provide the most reliable measure of propagation propensity and corresponding mechanical properties, the procedure is time consuming. However, this disadvantage can only be outweighed, if modelled metrics of instability become more reliable. This will require further validation studies – best performed by comprehensive field measurements in study plots equipped with an automatic weather station, and possibly an enhancement of the representation of mechanical properties in the model based on cold laboratory studies.

**Acknowledgments**

We are grateful to Franziska Zahner and Achille Capelli for help with the field work. Frank Techel provided the avalanche activity data. We thank Karl Birkeland and an anonymous reviewer for their constructive comments that helped to improve the paper. B. Reuter was supported by a grant of the Swiss National Science Foundation (200021_144392) and J. Gaume by an Ambizione grant of the Swiss National Science Foundation (PZ00P2_161329).

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

**Tables**

**Table 1: Snowpack characteristics and snow instability test results on the eight measurements days. For the PST the total number of tests and the number of tests with crack propagation to the end of the column, for the CT the score and the fracture type (SP: sudden planar, SC: sudden collapse), and for the ECT the fracture type (X: no fracture, N: initiation, but no propagation, P: propagation to column end) and the score are given (Greene et al., 2010).**

| Date | Snow depth (cm) | Slab thickness (cm) | Slab density (kg m$^{-3}$) | Load (kPa) | Test results | | |
|------|-------|-----------|---------|------|-----------|-----|------|
| | | | | | PST total/END | CT | ECT |
| 06 Jan 2015 | 115 | 59 | 245 | 1.42 | 5/2 | 19 SP | X, N20 |
| 14 Jan 2015 | 110 | 56 | 272 | 1.50 | 5/0 | 19 SC, 21 SC | P21 |
| 21 Jan 2015 | 126 | 72 | 275 | 1.94 | 2/1 | 13 SC | P21 |
| 28 Jan 2015 | 161 | 108 | 207 | 2.20 | 3/3 | 29 SC | X, P27 |
| 03 Feb 2015 | 172 | 117 | 242 | 2.78 | 3/3 | 22 SC | P27 |
| 13 Feb 2015 | 139 | 97 | 309 | 2.94 | 4/4 | 28 SP | P22 |
| 19 Feb 2015 | 147 | n/a | n/a | n/a | n/a | n/a | n/a |
| 03 Mar 2015 | 193 | 148 | 269 | 3.90 | 3/3 | 26 SC | P30, P30 |

**Figures**

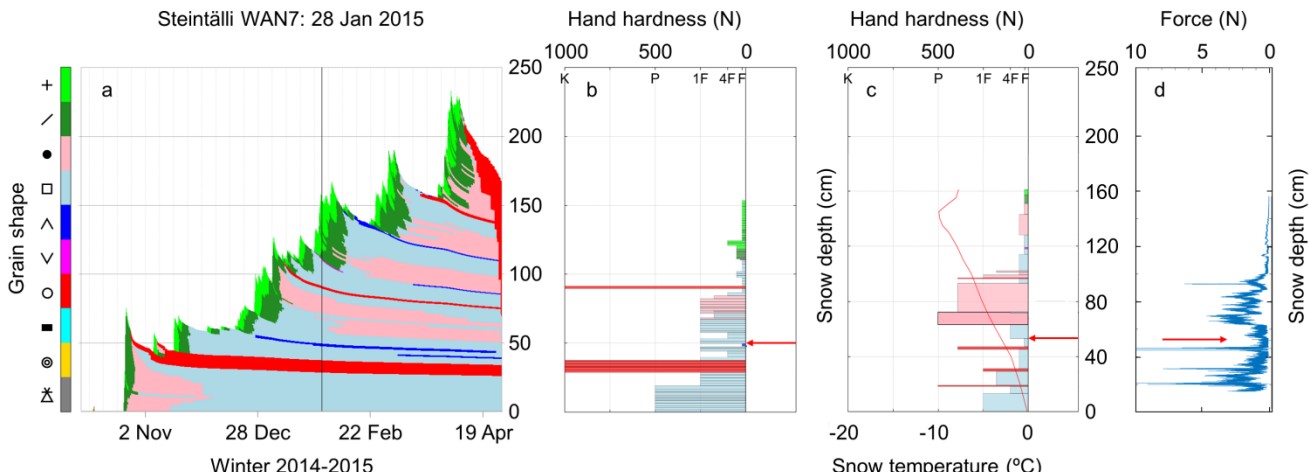

**Figure 1: (a) SNOWPACK simulation for the location of the automatic weather station (AWS) WAN7 for winter 2014-2015 showing the evolution of grain shape, black vertical line indicates date of snow profile (28 Jan 2015), (b) simulated snow profile for 28 Jan 2015, (c) manually observed snow profile at the location of the AWS on 28 Jan 2015, (d) corresponding SMP penetration force signal measured at the location of the manual profile. Red arrows point to the weak layer.**

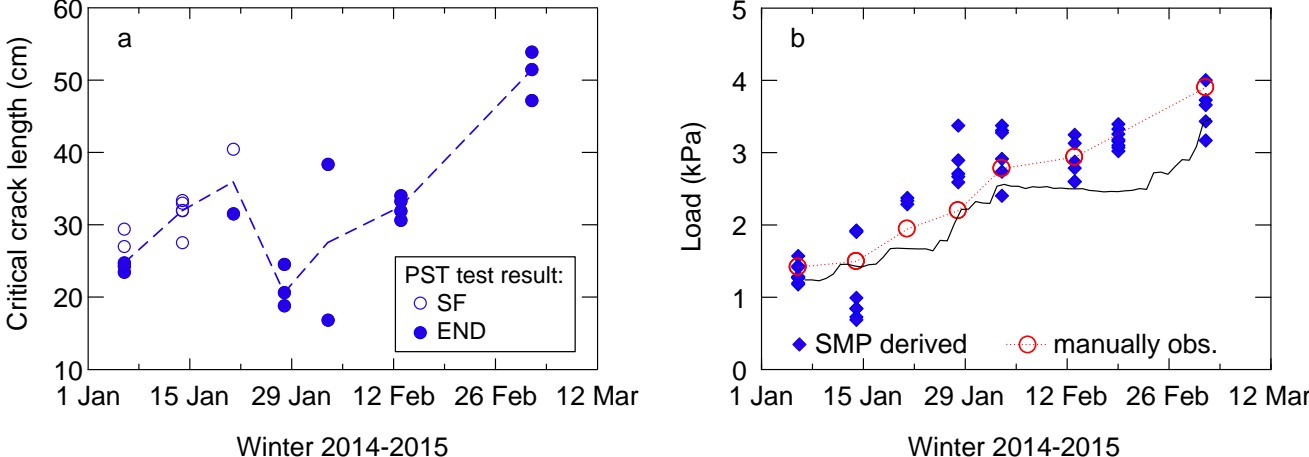

**Figure 2: (a) Critical cut length as observed in propagation saw tests (Number of cases: $N = 23$). Full circles indicate crack propagation to the very end of the column (END), open circles indicate partial propagation resulting in a fracture across the slab (SF); dashed line connects the median values per day. (b) Load on the weak layer: red open circles (connected by dotted line) show the load as calculated from the manually observed density and layer thickness; blue diamonds are corresponding SMP-derived values ($N = 50$). The black solid line indicates the load as provided by SNOWPACK.**

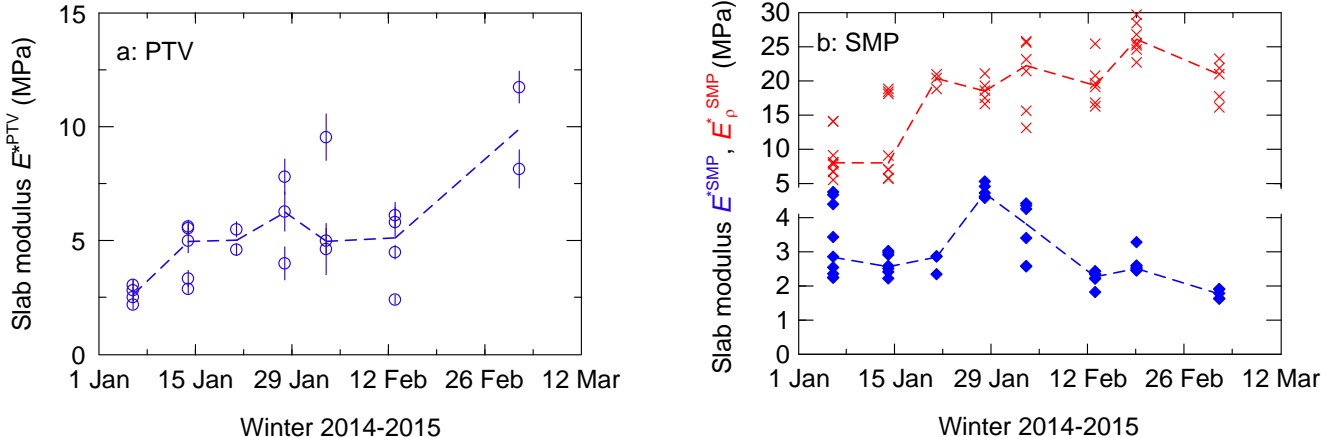

**Figure 3: Effective elastic modulus of the slab derived from (a) the bending of the slab via PTV analysis with vertical lines denoting measurement uncertainty ($N = 24$), and (b) from the SMP signal analysis ($N = 50$), either directly $E^{*SMP}$ (solid blue triangles) or via density $E^*_\rho{}^{SMP}$ (red crosses). Dashed lines connect the median values per day.**

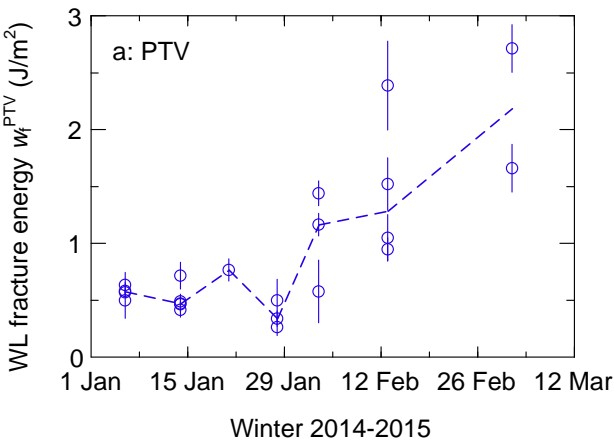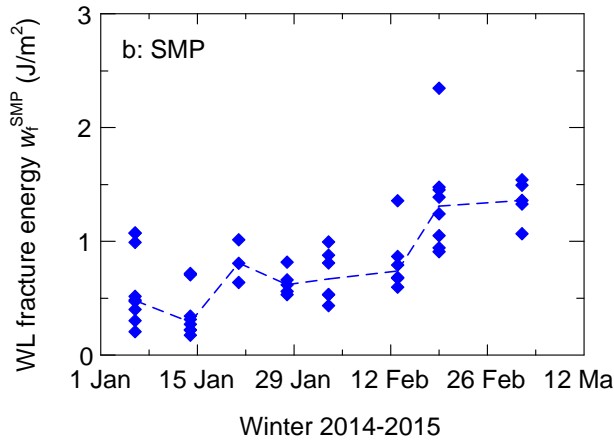

**Figure 4: Weak layer specific fracture energy derived from (a) the bending of the slab via PTV analysis ($N = 24$) with vertical lines denoting measurement uncertainty, and (b) from the SMP signal analysis ($N = 50$). Dashed lines connect the median values per day.**

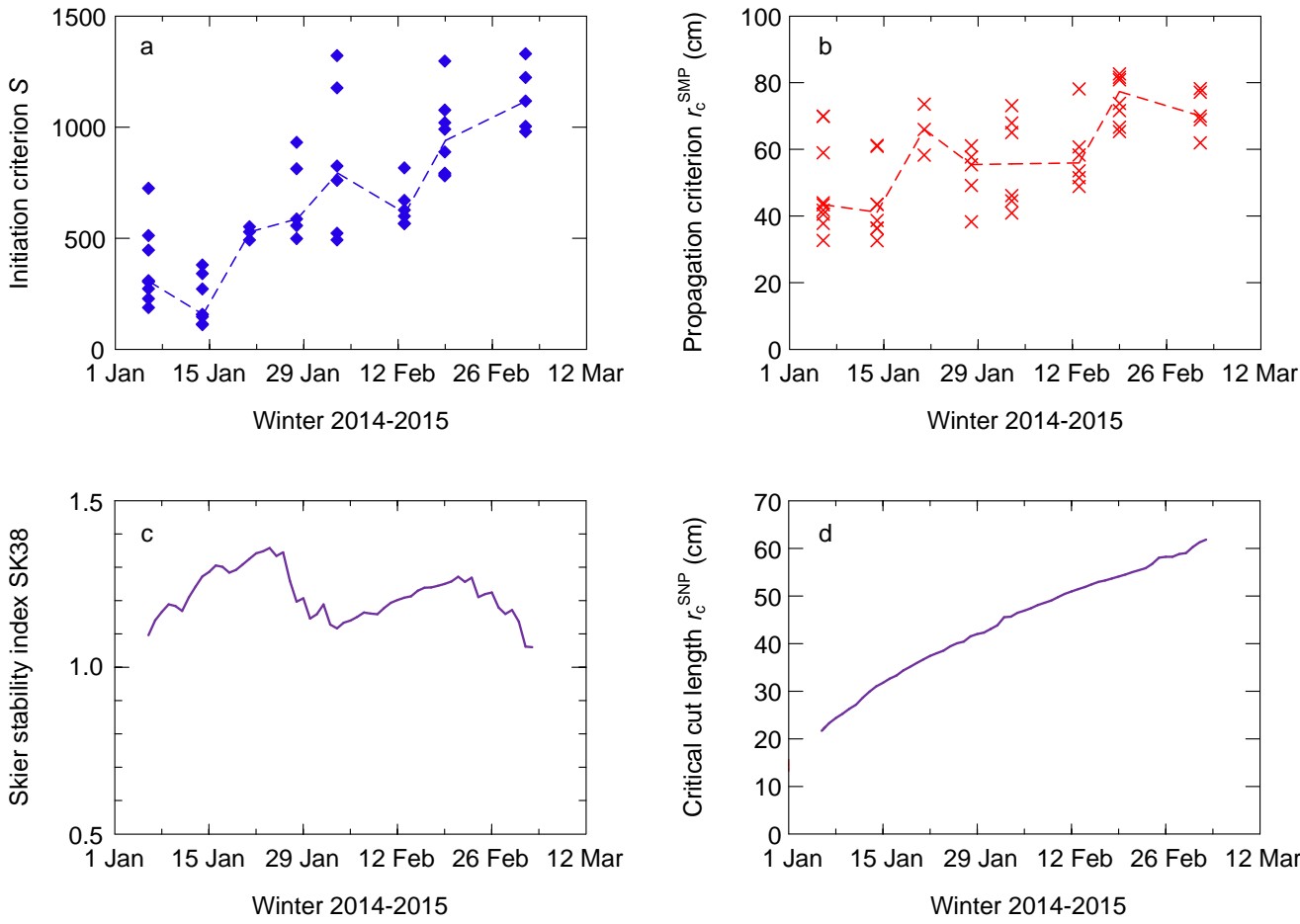

**Figure 5: Instability criteria: (a) SMP-derived failure initiation criterion *S*, and (b) SMP-derived crack propagation criterion $r_c^{SMP}$; dashed lines connect median values per day ($N = 50$). Output of the numerical snow cover model SNOWPACK: (c) Skier stability index SK38, and (d) modelled critical cut length $r_c^{SNP}$.**

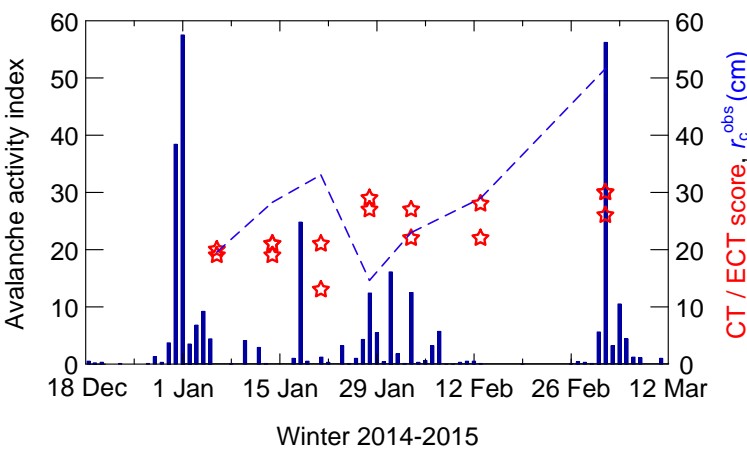

**Figure 6: Avalanche activity index for the region of Davos (columns), results of the CTs and ECTs performed concurrently with the snow profile observations on seven out of eight sampling days (red asterisks), the number of taps (score) is shown, and the observed critical cut length $r_c^{\mathrm{obs}}$ (dashed blue line, as in Fig. 2).**

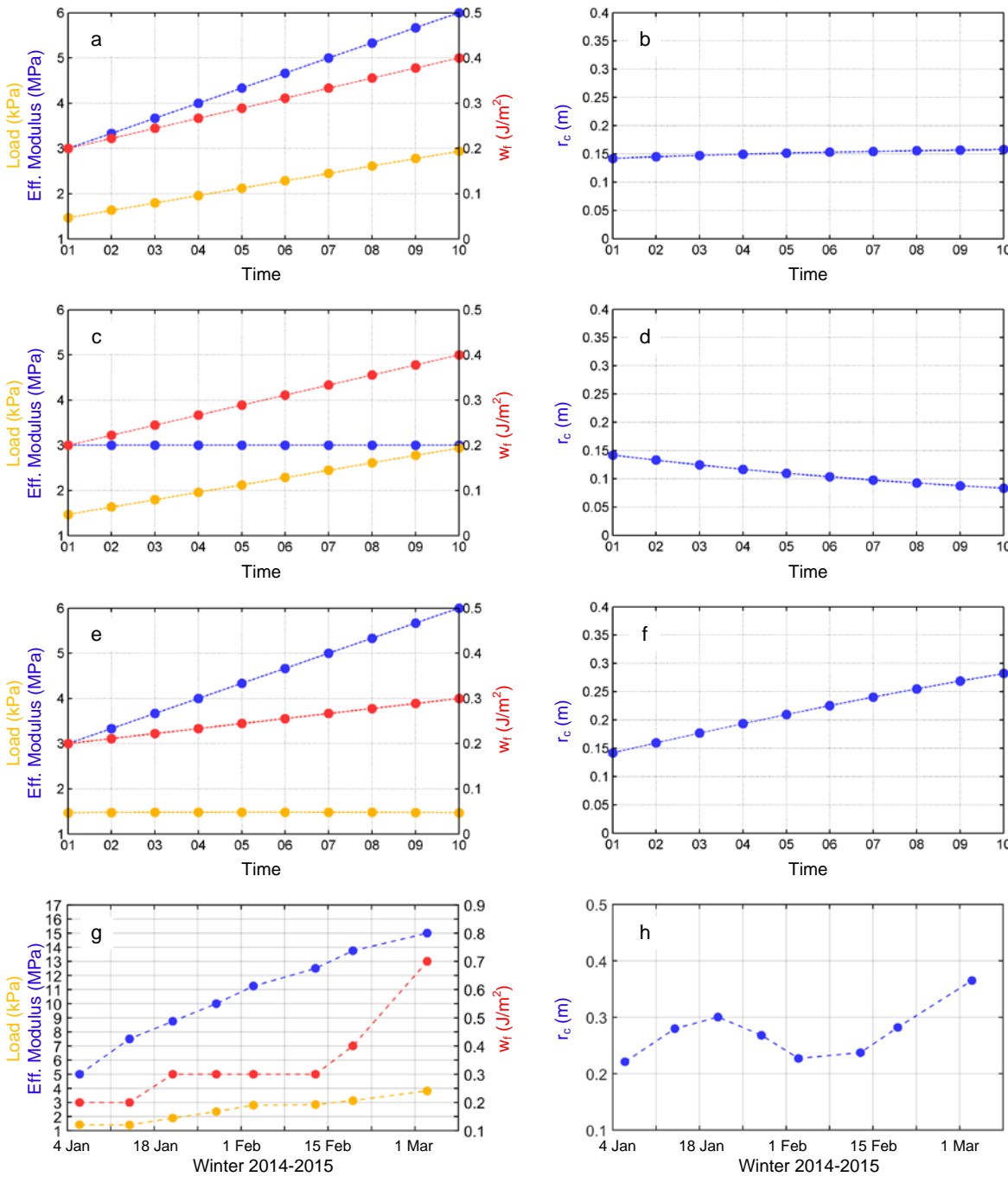

**Figure 7: Sensitivity study on how the critical cut length $r_c$ varies as a function of the load and the effective modulus of the slab, and the specific fracture energy of the weak layer $w_f$. Arbitrary units of time for the three simplified scenarios shown in panels (a) to (f). In the last scenario (g,h), the situation during the sampling period is supposed to be roughly mimicked.**