# Peer review of "Temporal evolution of crack propagation propensity in snow in relation to slab and weak layer properties"

_The Cryosphere, 2016_

## Referee Comment (RC1) · Anonymous Referee #1 · 31 May 2016

SUMMARY:

The authors monitored the temporal evolution of a {weak layer, slab} system during winter 2014-2015 in a field site located next to Davos. Typically, each week between 6 January 2015 and 3 March 2015 (8 days of measurements), they performed on the same site located next to an automatic weather station:

- three propagation saw test (PST) on which they measured the critical crack length, the full or partial crack propagation and the slab displacement field (PIV measurements),

- around five SMP profiles,

- a classical manual snow profile with a density profile

- CT/ECT tests.

The authors try to explain the observed temporal evolution of the PST critical crack length (general increase with a minimum the 28 January) by investigating the evolution of individual mechanical parameters of the weak layer and slab, namely the load on the weak layer, the weak layer fracture energy and the so-called bulk elastic modulus; and their interaction through the anti-crack model. They used previously developed methods to access these parameters from the measured data. They also used the SNOWPACK model to compute the critical length from the simulated snow profile with meteorological forcings from the automatic weather station. The authors show that monitoring the evolution of individual parameters cannot explain the observed critical crack length trend but that it is necessary to account for the complex interaction between these mechanical variables. The SMP metric is not able to reproduce the observed critical crack length. The SNOWPACK metric shows also an increase of the critical crack length.

GENERAL COMMENTS

The dataset collected by the authors is very interesting combining quantitative stability analysis (PST critical crack length) and highly resolved vertical hardness profile (SMP). Some of the results are of clear interest to the snow and avalanche community: the authors showed that both slab properties and weak layer cannot be individually monitored to understand the crack propagation propensity evolution; they also show that the previously developed SMP stability metric is not capable of capturing the evolution of the critical crack length. However, the methods are not well presented and appear as a black boxes where explanations on the basic assumptions are missing and the methods are mixed without an apparent logic. In particular, the SMP stability metric presentation is not clear in this form. Evaluating the stability metric of SNOWPACK from a modeled snow profile without showing that the modeled snowpack profile has something in common with the observations is not informative. The sensitivity analysis on a three parameters analytic function is based on four single cases. The trend analysis gives too much importance to a single day case that might be not statistically

representative.

Therefore, I recommend major revisions before publication.

MAJOR COMMENTS:

1) The dataset collected by the authors is very valuable. Indeed, the authors present it as the first comprehensive time series of a {weak layer, slab} system. It uses state-of-the-art measuring techniques (SMP, PST) combined with "traditional" measurements (manual stratigraphy and density, CT/ECT). Since one of the objective and strength of the paper is this dataset, it appears logical to provide this dataset as supplementary files (Caaml file for stratigraphy, stability tests, text file for SMP and avi file for PST videos).

2) The writing style on the mechanical background is often unscientific and requires precision and consistency. I have listed some of these problems:

- about the elastic modulus. You used the following terms without proper definition: "elastic modulus", "bulk modulus", "modulus", "effective modulus", "bulk effective modulus", "micro-mechanical modulus", "slab modulus", "stiffness", "elastic modulus with non-elastic parts of deformation". This vocabulary is misleading and is not suited for a scientific paper, where the mechanical concepts behind the used model should be precisely presented, which can be done in a simple way accessible to the snow community.

- you use the terms "propagation propensity", "propagation criterion r_c_SMP" ,"critical crack length", "propagation propensity metric", "crack propagation propensity" to refer to the same parameter r_c, or maybe not but this is not clear. Why don't you use consistently the well-defined "critical crack length" and explain only in the introduction that the critical crack length is an indicator of the more general concept of crack propagation propensity?

- "initiation probability", "initiation propensity", "initiation criterion", "initiation indices",

"skier stability index" ...

- delete vague and unspecific claims "reliable", "reliable in general", "distinct pattern", "relevant mechanical properties", "other mechanical properties"

3) It is hard to follow the history of the {weak layer, slab} system. It is necessary to add a one-page figure with eight sub-figures (one for each day of measurements) showing the manual stratigraphy (at least snow type and density), a SMP profile and the position of the weak layer.

4) In Heierli's model, the total mechanical energy of a PST crack of length r is composed of two terms: $V(r) = w_f * r + V_m(r)$ where $w_f * r$ is the weak layer fracture energy and $V_m(r)$ accounts for elastic deformation energy and changes in gravity potential energy of the slab. In case of a uniform slab, $V_m(r)$ can be computed analytically knowing the density, thickness and elastic modulus of the slab. In case of a FE model of a multilayer slab (density, thickness and elastic modulus per layer known), $V_m(r)$ can be calculated numerically. This is done for the SMP analysis. In case of a measured displacement/deformation field of the PST tests, $V_m(r)$ can also be calculated. This is done in the PST analysis. In both cases (SMP, PST method), the calculated $V_m(r)$ is used to fit the analytic mono-layer solution. The fitted analytic solution is then differentiated to obtain the critical crack length knowing the weak layer fracture energy (SMP method) or the weak layer fracture energy knowing the critical crack length (PST method). I don't understand why the $dV_m(r)/dr$ is not computed directly from the calculated $V_m(r)$ (or with smoothing of $V_m(r)$). This is not explained in the proposed references (Reuter et al, 2015 or van Herwijnen and Heierli, 2010). The bulk elastic modulus is a fitting parameter and it is unclear how physically-relevant it is. There is no clear reason why $V_m(r)$ on layered material should fit directly the mono-layer analytic solution. Provide a proper explanation and discussion on that. Moreover, recall the main hypothesis (elastic linear, only the slab contributes to deformation energy) of Heierli's model.

[Figure]

5) Section 2.4 describing the SMP signal processing is vague and unscientific. Many critical details are missing. It does not allow the reader to reproduce the presented method and appears as a black box. It requires a deep rewriting. It mixes method using different concepts that measures the same things differently e.g. Johnson and Schneebeli (1999) and shot-noise model used by Proksch, 2015. The window size for analysis, the SMP version, the adjustment parameters of (Proksch et al, 2015, calculated on a few alpine snow samples), the finite element layer mesh, etc. are missing. There is additional linear scaling with no convincing explanation. The calculation of layer Young's modulus from SMP elementary failure element is known to be poor and is inconsistent with the one based on density (Scapozza, 2004) used by the snow cover modeling (p5 l30). The failure initiation criterion S is not detailed and it is hard to notice that it does not incorporate snow load in comparison to SK38 which does, ... The reference to other papers is far from being sufficient and clear explanations won't take more than 30 lines.

6) The authors used the snow cover model forced by a nearby automatic weather station as an input of a new critical crack length estimator (Gaume et al. 2014a, 2016). Without any clue on how close the snowpack simulation to the observed snowpack, it is impossible to exploit the results of this analysis. It is well-known that one point evaluation of a snow cover model on stability criterion is difficult. Note that the only variables missing in Eq. (1) is the weak layer strength that could be fitted to get r_c_snp = r_c_obs, similarly to what is done for the PST.

Additionally, it is not clear to me how the avalanche activity index (concerning the area all around Davos ?) can help to analyze the measurement done in this particular site.

7) The pattern of the PST critical crack length is a general increase with a local minimum for one measurement day (28 January). As discussed (p6 l20-23, p10 l3-6), the spatial variability can significantly affect the stability even a few meters away. Given the poor representativity of one day of measurement to define a trend, and potential spatial variability, it would be reasonable when speaking of trend to not focus on the

minimum observed the 28 January but on the general trend (continuous increase of r_c). Note that this does not challenge the fact that the SMP should reproduce the same trend (since measured a few cm away from the PST); but the comparison with SNOWPACK is challenged. The explanations "we deem it unlikely that the observed pattern is entirely the result of spatial variability and does not reflect the temporal evolution", "Previous studies performed in level study plots have shown that measurements in general are reliable and that the effect of spatial variations is relatively small" are not convincing, at least in this form.

8) The sensitivity analysis is poor and based on four different cases. To my opinion, this cannot be called a sensitivity analysis. Differentiating Eq. (2) with respect to E, sigma and wf provides a way to perform this sensitivity analysis properly.

Note that the general comments are general and require re-wording of several parts of the paper and additional explanations, and not only taking into account specific minor points listed below.

MINOR COMMENTS:

abstract: the following terms are too vague : "distinct pattern", "other mechanical properties" "some of the relevant mechanical properties"

p1 l25: "how much stress due to a skier is transferred". Misleading sentence. All the stress is transferred to the ground. But it is distributed on a larger surface. Reword.

p1 l28: "with respect to the weak layer, a snowpack a weakness is" -> "the weak layer is"

p2 l2: "conceptual model". Describe this model in a few words.

p2 l7: "though the strengthening may lag behind the loading". Sound unscientific. Delete.

p2, l27: References to the model Surfex-Crocus (Vionnet, V. et al. Model Development

The detailed snowpack scheme Crocus and its implementation in SURFEX v7 . 2. Geoscientific Model Development 5, 773–791 (2012)) and Mepra (e.g. 1. Giraud, G. MEPRA an expert system for avalanche risk forecasting. in International Snow Science Workshop 97–104 (1992)) are clearly missing.

p3 Section 2.1: Is the snowpack completely dry during measurement period?

p4 l1-2: "The weak layer . . . December 2014". Explain how you know that.

p4 l2-3: "While no fracture . . . January 2015". I don't understand. Reword.

p4 l7: "The manual snow profile served as a reference". Do you mean that you performed manual stratigraphic matching to adjust the other snow profiles to the manual profile?

p4 l10: "at least three PST". It appears from Figure 1a) that there two other dates where less PST were performed.

p4 l14: "we cut the layer of faceted crystals at its upper interface". One of the main difficulty of the PST is to follow the weak layer of interest. As explained in Section 2.1, there was another FC layer just above the weak layer of interest. Showing the SMP profiles (see main comments) could help the reader to evaluate the likelihood of deviation of the saw cut in the weak layer.

p4 l18: Give version of SMP.

p4 l25: "the displacement of the markers was used to estimate the mechanical energy $V_m(r)$ with increasing crack length". As far as I understand, at this step, you also need the load, i.e. the density of the manual profile. Add explanation if this is correct.

p4 section 2.3: The critical crack length of the modeled PST is inherently equal (or very close) to the observed critical crack length since the observation is used to fit w_f. This might not appear clearly to the reader. Please add this kind of explanation.

p5 l28: "the shear modulus of the weak layer which was estimated". How ?

p5 l30: I suggest to explicitly indicate the power law relation used here.

p6 Eq2: To my opinion, this equation in this form does not give any information to the reader. Delete or give detail on all terms.

p6 l23-26: "By then, the weak layer of ... resulting in a load of almost 4 kPa." Belong to the load section 3.2?

p7 l29: "0.3 J m -2 to about 1.5 J m -2". Recall that this range results from a linear scaling between w_f_SMP and w_f_PTV.

p8 l3: S = shear_strength / skier stress should be described in Methods. Adding two lines of description is not a big deal and would clarify the message. See main comments.

p8 l10: SK38 = shear_strength / (skier stress + weight_stress) should be described in Methods. See main comments.

p8 l22-24: The CT/ECT tests could be better used to evaluate the initiation criteria (SMP, SK38).

p9 l14-18: I don't understand this paragraph. The rc_obs is used to compute w_f_PTV. That w_f_PTV as input in Heierli's model gives the same trend for r_c does not appear to me as a finding ??? Clarify.

p9 l27-28: "Only when the load had reached 2 kPa, all cracks fully propagated towards the end of the column. This finding suggests that the slab was initially not strong enough to support the propagation". I don't understand the logic link between these two sentences (load/strength ?). Clarify.

p10 l7 "5.9 cm". This is not a range.

p10 l15-19: "The errors associated with the parameters ... the dots in the PTV analysis)." This a new info that belongs to Methods and Results sections.

p10 l19-22: Adding error-bars on the figures 2a, 3a would help to illustrate this discussion. Moreover, you might go further in this discussion. Indeed w_f depends only on one layer whereas E is an integrated value on the slab layers and might thus be less sensitive to the spatial variations of one layer.

p10 l26: "validated" -> "evaluated"

p11 l3: "is in line with the observations in particular when considering the CT and ECT scores.". What are the others ?

p11 l10: "– suggesting that the propagation propensity decreased". Delete

p11 l10-11: "This behavior follows from the fact that two of the essential variables, the bulk modulus and the weak layer shear strength also increase with time." From your sensitivity analysis (figures 6a,b) and the fact that you get the same results for Eq. (1), this is not a sufficient explanation.

p11 l14-15: "However, it seems premature to rate this metric as it has to be considered as being still in an experimental state." I agree this is a very valuable criterion to help to synthesize the data of snowpack models. However, the explanation is evasive. To my opinion, evaluation of this metric on one point stability observations with potential errors in meteorological forcing and SNOWPACK modeling is the main problem. See main comments. Delete or reword.

p11 l19: "The parameter most strongly influencing the critical cut length seems to be the load". Not shown in results. Can be quantified. See main comments.

Figures: what is the running median smoother (kernel size ?)

Figure 1: a) give r_c in m for consistency. b) indicate in the figure what is the black solid line.

---

## Referee Comment (RC2) · K. B. Birkeland (Referee) · 1 Jun 2016

This paper presents a unique dataset of temporal changes in crack propagation propensity over the course of a season, and how that propagation propensity related to temporal changes in the slab and weak layer. The authors utilized the latest tools for their work, including analyzing high speed video with PTV, making measurements with the SMP, and modelling the evolution of the snowcover with SNOWPACK. The paper is a valuable addition to the literature, and I believe it should be published after it is revised.

My first suggestion is that the authors consider a different title. Since the paper really focuses on crack propagation propensity, the title should better reflect that. Perhaps something along the lines of "Temporal evolution of crack propagation propensity in

view of slab and weak layer properties" or similar? Or, even more specifically, "Temporal evolution of critical cut lengths. . ."?

Also, it would be nice if the authors could briefly describe more of the methods used. I know that they will not want to repeat long sections of previous work, but if it would be useful for the reader if they could provide even a few more details about some of the SMP and SNOWPACK derived parameters. More background information will help the reader better assess those parameters and how they performed.

Major comments:

One primary concern about the paper has to do with Figure 1a and the evolution of the cut length of the PSTs. In this graph it appears that the authors are mixing results that go to END with result that are SFs. Can the authors discuss and defend why they feel this is an appropriate treatment of these data? In my experience I have seen situations where SFs have longer cut lengths, but then as the PSTs transition to END results the cut lengths decrease. At this point, I am not convinced that you can treat the two sets of results (END and SFs) as the same and show a temporal trend with them. I would suggest that they defend this, or they only consider the tests that went to END.

Another primary concern about this paper is that I feel a much more robust discussion of the results is warranted. The authors have presented many interesting results, both in terms of the temporal evolution of various parameters and in the comparison of different methods of tracking those temporal changes (between the field tests, PTV, the SMP, and SNOWPACK). However, in my opinion the authors do not fully discuss many of these findings. Some examples:

- The temporal changes of effective elastic modulus of the slab derived from PTV and derived from the SMP do not match (Figure 2). However, this discrepancy is not discussed. Which one of these two techniques do the authors believe is closer to capturing the "true" change in the elastic modulus? It seems to me that the PTV results more closely align with changes I've observed in the field. If this is the case for these

data, can the author suggest ways the SMP techniques can be improved? - The SMP's derived critical cut length did not match the observed changes in the PST. Why do the authors think this is the case? Is this some shortcoming in the SMP technique, or are the data presented in this paper somehow different from the data used to develop the SMP derived critical cut length? Does this finding shed additional uncertainty on the SMP derived cut length? - On page 12, line 10 the authors state that this metric is experimental so it is premature to rate it. I disagree with this statement. If the metric is seen as useful enough to be included in the paper, then I feel that it is appropriate to fully evaluate it and rate its usefulness. - Another point that is not fully discussed is the difference between the elastic modulus values calculated using PTV and those calculated with the SMP. It would be nice to have a paragraph discussing these differences, why they occur, and whether there are ways to get better measurements out of some of the techniques. This could be placed after the paragraph ending on Page 11, Line 19. Looking at Figure 2, the numbers for the SMP seem strange (staying the same or even going down over the season), while the numbers for PTV seem more realistic. What do the authors think about this and how might they explain it?

Minor comments:

- Most figures feature a dashed line that is a "running median smoother". It would be helpful to know how the authors calculate this smoother. Also, are the cut lengths in Figure 1a treated the same whether the test went to END or was SF? It appears they were, but the authors may wish to state that in the text. - Page 2, Line 1. It is true that Sigrist and Schweizer (2007) were "among the first to emphasize the importance of the slab layers and weak layers", but there were others that emphasized that point either prior to, or at the same time as, 2007. Those include the MS thesis by B.C. Johnson (2001), the paper by Johnson and Jamieson (2004 in CRST), the PhD dissertation by van Herwijnen (2005), the paper by van Herwijnen and Jamieson (2007 in CRST), and the thesis by Gauthier (2007). Since this has been an important point, I'd encourage the authors to add some of those other publications to this citation. Other earlier work

also talks about the slab, but in terms of "emphasizing" the slab, it really began to be more clearly stated in the 2000s with the work by Johnson, Jamieson, van Herwijnen, Sigrist and Schweizer. - Page 2, line 18 and 19. Do the authors believe that the "shear strength of the weak layer is important for failure initiation" in the case of a triggered avalanche from flat terrain? - Page 7, line 6. When the authors state that "cracks did not always fully propagate", it would be useful if they stated how many tests were done and how many propagated (i.e., something along the lines of "when we did the first PSTs, cracks fully propagated in two of five tests, while slab fractures occurred in the other three tests" or whatever the numbers were). - Page 7, line 8. Like above, it would be nicer to know the number of tests instead of just writing "all tests". - Page 7, line 18. It seems to me that the data demonstrate that the propagation propensity decreased definitively (rather than slightly) between the first two days because on the second day all of the tests were SF while on the first day there were some that went to END. This could be due to the shallower nature of the snow in that part of the plot, as discussed by the authors, or it might have to do with a change in the slab. I have observed a decrease in propagation propensity (from more END results to more SF results) when a slab loses tensile strength due to near-surface faceting. - Page 9, line 15 and Figure 5. Did all of the ECTs propagate across the column (ECTPs) or did some not (ECTNs)? That should be made clear here or on Figure 5.

Typographical/grammatical errors:

- Page 1, Line 8, add "(PSTs)" after "propagation saw tests" since "PST" is used later in the abstract. - Page 1, Line 21, delete "considering the slab," - Page 1, Line 32, replace "but" with "and" - Page 2, line 14 and 15. It seems the information for those two sentences comes from a single reference, but the authors cite both Jamieson and Schweizer, 2000 and Schweizer et al., 1998. Which reference is correct? - Page 3, line 3, delete "exists" - Page 4, line 13, add an "s" to "PST" so it reads "PSTs" - Page 5, line 3, delete the comma that is after "modulus" - Page 7, line 9. You cannot have two semicolons in the same sentence. You will need to re-word to remove one of them.

---

## Author Comment (AC1) · 29 Jul 2016

**Reply to Referee #1**

We thank referee #1 for the thorough review and believe that these comments will be very helpful for preparing an improved revised manuscript. In the following we reply to the comments in detail and describe the changes we intend to make in the revised manuscript.

**SUMMARY:**

The authors monitored the temporal evolution of a weak layer-slab system during winter 2014-2015 in a field site located next to Davos. Typically, each week between 6 January 2015 and 3 March 2015 (8 days of measurements), they performed on the same site located next to an automatic weather station:

- three propagation saw test (PST) on which they measured the critical crack length, the full or partial crack propagation and the slab displacement field (PIV measurements),
- around five SMP profiles,
- a classical manual snow profile with a density profile
- CT/ECT tests.

The authors try to explain the observed temporal evolution of the PST critical crack length (general increase with a minimum the 28 January) by investigating the evolution of individual mechanical parameters of the weak layer and slab, namely the load on the weak layer, the weak layer fracture energy and the so-called bulk elastic modulus; and their interaction through the anti-crack model. They used previously developed methods to access these parameters from the measured data. They also used the SNOWPACK model to compute the critical length from the simulated snow profile with meteorological forcings from the automatic weather station. The authors show that monitoring the evolution of individual parameters cannot explain the observed critical crack length trend but that it is necessary to account for the complex interaction between these mechanical variables. The SMP metric is not able to reproduce the observed critical crack length. The SNOWPACK metric shows also an increase of the critical crack length.

**GENERAL COMMENTS**

The dataset collected by the authors is very interesting combining quantitative stability analysis (PST critical crack length) and highly resolved vertical hardness profile (SMP). Some of the results are of clear interest to the snow and avalanche community: the authors showed that both slab properties and weak layer cannot be individually monitored to understand the crack propagation propensity evolution; they also show that the previously developed SMP stability metric is not capable of capturing the evolution of the critical crack length. However, the methods are not well presented and appear as a black boxes where explanations on the basic assumptions are missing and the methods are mixed without an apparent logic. In particular, the SMP stability metric presentation is not clear in this form. Evaluating the stability metric of SNOWPACK from a modeled snow profile without showing that the modeled snowpack profile has something in common with the observations is not informative. The sensitivity analysis on a three parameters analytic function is based on four single cases. The trend analysis gives too much importance to a single day case that might be not statistically representative. Therefore, I recommend major revisions before publication.

- We agree that our description of the methods is minimal mainly referring to previous work. We will change this approach and provide more details on each of the methods we use.
- We will provide information on the SNOWPACK simulations so that the reader can assess whether the simulated stratigraphy has something in common with the observations (see below).

- The sensitivity analysis is exemplary and focusses on the temporal evolution. This paragraph was meant to illustrate how the various factors interact. We will change the title as it is obviously not a complete sensitivity analysis.
- With regard to the temporal evolution and the observed minimal values towards the end of January, we will discuss the representativity more thoroughly. We would like to point out that minimal values of snow instability tests are in general more trustworthy. In the case of the propagation saw test, any measurement and observation errors increase the cut length. Low values of the cut length therefore almost always represent the real conditions.

**MAJOR COMMENTS:**

1) The dataset collected by the authors is very valuable. Indeed, the authors present it as the first comprehensive time series of a weak layer, slab system. It uses state-of-the-art measuring techniques (SMP, PST) combined with "traditional" measurements (manual stratigraphy and density, CT/ECT). Since one of the objective and strength of the paper is this dataset, it appears logical to provide this dataset as supplementary files (Caaml file for stratigraphy, stability tests, text file for SMP and avi file for PST videos).

Whereas it is generally a good idea to provide the data, we believe sharing the data is not straightforward. We are not dealing with 'simple' weather data, but with data from various sources (SMP, PTV, SNOWPACK, profiles), which then have to be processed to get to the results. Furthermore, the processing is not trivial. Hence, we rather prefer to provide some of the data in the supplementary material. In particular, we will provide the manual profiles including an SMP profile for all days in the supplement and add a figure to the main text showing the SNOWPACK simulation (see below). And of course, we will provide the data on request to others who like to collaborate.

Figure: (a) SNOWPACK simulation for the location of the automatic weather station (AWS) WAN7 for winter 2014-2015 showing the evolution of grain size, (b) simulated snow profile for 28 Jan 2015, (c) manually observed snow profile at the location of the AWS on 28 Jan 2015. Arrows denote the weak layer.

2) The writing style on the mechanical background is often unscientific and requires precision and consistency. I have listed some of these problems:

- about the elastic modulus. You used the following terms without proper definition:"elastic modulus", "bulk modulus", "modulus", "effective modulus", "bulk effective modulus", "micro-mechanical modulus", "slab modulus", "stiffness", "elastic modulus with non-elastic parts of deformation". This vocabulary is misleading and is not suited for a scientific paper, where the mechanical concepts behind the used model should be precisely presented, which can be done in a simple way accessible to the snow community.

We acknowledge that the vocabulary on the deformation behavior of the slab may be hard to follow and will carefully revise the manuscript to account for this issue. The problem arises from the fact that model assumptions, e.g. linear elasticity, do not fit what is actually observed and can be measured in the field; in addition, the slab is layered and not uniform. Therefore, there is some need for specific terms and it is not sufficient to just talk about the modulus. For example, the modelling approach by Heierli et al. (2008) includes the elastic modulus (Young's modulus), what we measure with PTV is an effective bulk modulus (bulk because layering is disregarded, effective because it includes not only purely elastic parts of deformation), what is derived from the SMP is the micro-mechanical modulus. We will explain these subtleties in detail in the revised manuscript.

- you use the terms "propagation propensity", "propagation criterion r\_c\_SMP", "critical crack length", "propagation propensity metric", "crack propagation propensity" to refer to the same parameter r\_c, or maybe not but this is not clear. Why don't you use consistently the well-defined "critical crack length" and explain only in the introduction that the critical crack length is an indicator of the more general concept of crack propagation propensity?
- "initiation probability", "initiation propensity", "initiation criterion", "initiation indices", "skier stability index" ...
- delete vague and unspecific claims "reliable", "reliable in general", "distinct pattern", "relevant mechanical properties", "other mechanical properties"

We will re-consider the use of the terms in connection with failure initiation and crack propagation. As we present measured as well as modelled values there is some need for distinction between the various measures.

We will thoroughly go through manuscript and remove vague and unspecific terms.

3) It is hard to follow the history of the weak layer-slab system. It is necessary to add a onepage figure with eight sub-figures (one for each day of measurements) showing the manual stratigraphy (at least snow type and density), a SMP profile and the position of the weak layer.

We will provide a figure showing the SNOWPACK simulation as well as modelled and observed profile for a specific day (see above). In addition, we will provide all the manually observed profiles including an SMP profile in the supplementary material.

4) In Heierli's model, the total mechanical energy of a PST crack of length r is composed of two terms:  $V(r) = w_f * r + Vm(r)$  where  $w_f * r$  is the weak layer fracture energy and Vm(r) accounts for elastic deformation energy and changes in gravity potential energy of the slab. In case of a uniform slab, Vm(r) can be computed analytically knowing the density, thickness and elastic modulus of the slab. In case of a FE model of a multilayer slab (density, thickness and elastic modulus per layer known), Vm(r) can be calculated numerically. This is done for the SMP analysis. In case of a measured displacement/deformation field of the PST tests, Vm(r) can also be calculated.

This is done in the PST analysis. In both cases (SMP, PST method), the calculated Vm(r) is used to fit the analytic mono-layer solution. The fitted analytic solution is then differentiated to obtain the critical crack length knowing the weak layer fracture energy (SMP method) or the

weak layer fracture energy knowing the critical crack length(PST method). I don't understand why the dVm(r)/dr is not computed directly from the calculated Vm(r)(or with smoothing of Vm(r)). This is not explained in the proposed references (Reuter et al, 2015 or van Herwijnen and Heierli, 2010). The bulk elastic modulus is a fitting parameter and it is unclear how physically-relevant it is. There is no clear reason why Vm(r) on layered material should fit directly the mono-layer analytic solution. Provide a proper explanation and discussion on that. Moreover, recall the main hypothesis (elastic linear, only the slab contributes to deformation energy) of Heierli's model.

We will provide a proper explanation, and also refer to the recent paper by van Herwijnen et al. (2016) where the PTV method is now explained in detail. We will use their refined approach, i.e. the adjusted mechanical energy to account for differences between the model of Heierli et al. (2008) and the FE simulations.

Taking the derivative of the raw data to derive  $w_f$  would not work, as there is too much scatter and this would result in very unreliable values of  $w_f$ .

We agree that the critical cut length can be computed with the FE model using the SMP slab properties and the SMP-derived specific fracture energy  $w_{\rm f}$ , but would require an iterative approach to find  $r_{\rm c}$ .

5) Section 2.4 describing the SMP signal processing is vague and unscientific. Many critical details are missing. It does not allow the reader to reproduce the presented method and appears as a black box. It requires a deep rewriting. It mixes method using different concepts that measures the same things differently e.g. Johnson and Schneebeli (1999) and shotnoise model used by Proksch, 2015. The window size for analysis, the SMP version, the adjustment parameters of (Proksch et al, 2015, calculated on a few alpine snow samples), the finite element layer mesh, etc. are missing.

There is additional linear scaling with no convincing explanation. The calculation of layer Young's modulus from SMP elementary failure element is known to be poor and is inconsistent with the one based on density (Scapozza, 2004) used by the snow cover modeling (p5 I30). The failure initiation criterion S is not detailed and it is hard to notice that it does not incorporate snow load in comparison to SK38 which does, ... The reference to other papers is far from being sufficient and clear explanations won't take more than 30 lines.

As mentioned above we will provide more details in general and in particular on the methods and not simply refer to previous work.

The additional linear scaling is simply introduced to obtain SMP-derived values that are comparable to PTV-derived values. This is clearly described as such. It is clear that SMP-derived values have some deficiencies, see Reuter et al. (2013), and we will discuss this more thoroughly in the revised manuscript. Furthermore, we will consider as an alternative approach, deriving density from the SMP signal, and then directly use a parameterization such as provided by Scapozza (2004) to obtain a modulus.

6) The authors used the snow cover model forced by a nearby automatic weather station as an input of a new critical crack length estimator (Gaume et al. 2014a, 2016). Without any clue on how close the snowpack simulation to the observed snowpack, it is impossible to exploit the results of this analysis. It is well-known that one point evaluation of a snow cover model on stability criterion is difficult. Note that the only variables missing in Eq. (1) is the weak layer strength that could be fitted to get  $r_c snp = r_c$  obs, similarly to what is done for the PST.

Additionally, it is not clear to me how the avalanche activity index (concerning the areaall around Davos?) can help to analyze the measurement done in this particular site.

The SNOWPACK simulation reproduced the snow stratigraphy reasonably well. We will provide the simulated stratigraphy in a new Figure (see above).

We certainly agree that stability predictions from simulated snow stratigraphy are challenging. We strongly believe that these stability predictions should be validated at the locations of automatic weather stations.

With regard to the comment on Eq. (1), we agree that the only missing variable is the weak layer strength, however, we are not sure we understand the reviewer's point. The shear strength cannot be determined form the measured critical cut length, otherwise the model would no longer be predictive. The shear strength is obtained from the parametrization implemented in the snow cover model SNOWPACK based on the work of Jamieson and Johnston (2001).

As we perform our measurements in a representative study plot commonly used in operational forecasting to extrapolate to the surrounding terrain (e.g., Gauthier et al., 2010; Jamieson et al., 2007), we added the avalanche activity data for comparison with the local stability evaluations.

7) The pattern of the PST critical crack length is a general increase with a local minimum for one measurement day (28 January). As discussed (p6 l20-23, p10 l3-6), the spatial variability can significantly affect the stability even a few meters away. Given the poor representativity of one day of measurement to define a trend, and potential spatial variability, it would be reasonable when speaking of trend to not focus on the minimum observed the 28 January but on the general trend (continuous increase ofr\_c). Note that this does not challenge the fact that the SMP should reproduce the same trend (since measured a few cm away from the PST); but the comparison with SNOWPACK is challenged. The explanations "we deem it unlikely that the observed pattern is entirely the result of spatial variability and does not reflect the temporal evolution", "Previous studies performed in level study plots have shown that measurements in general are reliable and that the effect of spatial variations is relatively small" are not convincing, at least in this form.

We will re-consider the local minimum that we observed at the end of January 2015. In fact, low critical cut lengths were not only observed on 28 January but also on 5 February. On 5 February there are only two measurements with a large difference between them. However, low PST results are in general more trustworthy than high ones, if they concurrently occur, since any error while performing the test will increase the cut length. Furthermore, on 28 January 2015, for the first time, all cracks propagated to the end of the PST column indicating that the crack propagation propensity had increased. Finally, the additional loading towards the end of January 2015 resulted in many avalanches and shooting cracks were frequently observed also indicating increased propagation propensity. We will re-assess the issue of measurement accuracy and spatial variability, reword the corresponding statements and clearly denote them as interpretation.

The sentence "*Previous studies performed in level study plots have shown that measurements in general are reliable and that the effect of spatial variations is relatively small*" is supported by two references to previous work just following this sentence (page 10, lines 11-14).

8) The sensitivity analysis is poor and based on four different cases. To my opinion, this cannot be called a sensitivity analysis. Differentiating Eq. (2) with respect to E, sigma and wf provides a way to perform this sensitivity analysis properly.

Note that the general comments are general and require re-wording of several parts of the paper and additional explanations, and not only taking into account specific minorpoints listed below.

The purpose of this paragraph is to illustrate how changes of the modulus, the load and the specific fracture energy with time will affect the temporal evolution of the critical cut length. We will no longer call this a sensitivity study, but select a new title for the paragraph: Case studies.

We agree, that differentiating Eq. (2) with respect to E,  $\sigma$  and  $w_f$  would reveal the dependence of the critical cut length for a single parameter. However, these dependencies, considered independently are obvious: the cut length decreases with increasing load, and increases with increasing slab modulus and weak layer fracture energy. However, their interplay in course of time cannot easily be assessed – and the four examples we provide simply show that entirely different evolutions are possible.

MINOR COMMENTS:

abstract: the following terms are too vague : "distinct pattern", "other mechanical properties""some of the relevant mechanical properties"

We will clarify the terms or remove them.

p1 l25: "how much stress due to a skier is transferred". Misleading sentence. All thestress is transferred to the ground. But it is distributed on a larger surface. Reword.

We will reword the sentence: "... the slab layers determine the magnitude of the stress due to a skier at the depth of the weak layer".

p1 l28: "with respect to the weak layer, a snowpack a weakness is" -> "the weak layeris"

We will reword the sentence as suggested.

p2 l2: "conceptual model". Describe this model in a few words.

We will add some explanations as suggested.

p2 17: "though the strengthening may lag behind the loading". Sound unscientific.Delete.

We will reword to: "... the rate of strength increase may lag behind the rate of loading."

p2, I27: References to the model Surfex-Crocus (Vionnet, V. et al. Model Development The detailed snowpack scheme Crocus and its implementation in SURFEX v7.2. Geoscientific Model Development 5, 773–791 (2012)) and Mepra (e.g. 1. Giraud, G.MEPRA an expert system for avalanche risk forecasting. in International Snow Science Workshop 97–104 (1992)) are clearly missing.

Thanks for pointing this out. We are certainly aware of the French model forecasting chain and rate it highly. However, it is unclear to us how the temporal evolution of strength is modeled, and in general how the strength is derived. We are not aware that in the various publications about MEPRA this is described in detail. As far as we know, in the paper by Vionnet et al. (2012) the words "strength" and "cohesion" do not even appear. In Giraud (1993) there are also no details given on how the strength is determined. If the reviewer can provide the reference where the function is described we are more than happy to include it.

p3 Section 2.1: Is the snowpack completely dry during measurement period?

Yes, the snowpack was completely dry – apart from some melting at the surface in early January resulting in a thin crust (see Figure above).

p4 I1-2: "The weak layer . . . December 2014". Explain how you know that.

We know as we closely follow the snowpack evolution and are in the field several times a week. This was the decisive weak layer at the end of December 2014. As mentioned on page 4, lines 2-3, there are no profiles available that were performed at fracture lines to support this assumption, but the particular weak layer consistently showed up as the primary failure layer in snow instability tests in the days following the avalanche cycle.

p4 I2-3: "While no fracture . . . January 2015". I don't understand. Reword.

See reply above. We will reword the sentence in the revised manuscript to clarify the argumentation.

*p4 I7: "The manual snow profile served as a reference". Do you mean that you performed-manual stratigraphic matching to adjust the other snow profiles to the manualprofile?*

The manual snow profile served as a reference to, for example, indicate the depth of the weak layer or other prominent layers.

*p4 I10: "at least three PST". It appears from Figure 1a) that there two other dates where less PST were performed.*

As mentioned on page 4, line 15 some test results had to be discarded since the cut was not performed consistently close to the interface which we only realized once we analyzed the videos. For that reason, we only have two test results on two days.

p4 I14: "we cut the layer of faceted crystals at its upper interface". One of the main difficulty of the PST is to follow the weak layer of interest. As explained in Section2.1, there was another FC layer just above the weak layer of interest. Showing the SMP profiles (see main comments) could help the reader to evaluate the likelihood of deviation of the saw cut in the weak layer.

As we filmed all tests we can easily assess whether the tests were properly performed – and have of course done so. As mentioned above, we will provide one SMP profile per measurement day in the supplementary material. However, the SMP profiles are less suited to assess a potential deviation while cutting the weak layer.

p4 I18: Give version of SMP.

We used SMP version 2.

p4 l25: "the displacement of the markers was used to estimate the mechanical energy Vm (r) with increasing crack length". As far as I understand, at this step, you also need the load, i.e. the density of the manual profile. Add explanation if this is correct.

Thanks for pointing this out; we will add that the density of the manual profile is used to evaluate the mechanical energy.

p4 section 2.3: The critical crack length of the modeled PST is inherently equal (or very close) to the observed critical crack length since the observation is used to fit w\_f. This might not appear clearly to the reader. Please add this kind of explanation.

The critical crack length is modelled from the weak layer fracture energy  $w_f$  as derived from the SMP. It is independent of the observed critical crack length. We will clarify this in the revised manuscript.

p5 l28: "the shear modulus of the weak layer which was estimated". How ?

Following Gaume et al. (2016) we used a constant value of the shear modulus  $G_{WL} = 0.2MPa$  according to the laboratory experiments performed on snow failure by Reiweger et al. (2010); for the Poisson's ratio of the slab we assumed a value of 0.2. We will reword this statement to: "..., for the shear modulus of the weak layer a constant value (0.2 MPa) was assumed, based on the laboratory experiments by Reiweger et al. (2010)."

p5 I30: I suggest to explicitly indicate the power law relation used here.

We will provide the relation as suggested.

p6 Eq2: To my opinion, this equation in this form does not give any information to the reader. Delete or give detail on all terms.

We will add the terms as suggested.

p6 I23-26: "By then, the weak layer of ... resulting in a load of almost 4 kPa." Belong to the load section 3.2?

This part of the sentence simply makes the link between slab thickness and density on one hand and load on the other hand so that the reader can better relate load values to commonly used parameters such as slab thickness.

p7 l29: "0.3 J m -2 to about 1.5 J m -2". Recall that this range results from a linear scaling between  $w_f$ \_SMP and  $w_f$ \_PTV.

We will mention the scaling here again.

p8 I3: S = shear\_strength / skier stress should be described in Methods. Adding two lines of description is not a big deal and would clarify the message. See main comments.

As mentioned above, we will introduce the SMP metrics in more detail in the Methods section of the revised manuscript.

*p8 I10:* SK38 = shear\_strength / (skier stress + weight\_stress) should be described in Methods. See main comments.

We will introduce the SK38 in more detail in the Methods section of the revised manuscript.

p8 I22-24: The CT/ECT tests could be better used to evaluate the initiation criteria(SMP, SK38).

Thanks for this suggestion; we will also discuss the CT/ECT results with respect to the initiation criteria in the revised manuscript.

p9 I14-18: I don't understand this paragraph. The rc\_obs is used to compute w\_f\_PTV. That w\_f\_PTV as input in Heierli's model gives the same trend for r\_c does not appear to me as a finding ??? Clarify.

This paragraph is to illustrate that under certain assumptions for the temporal evolution of *E*,  $\sigma$  and *w*f the critical cut length can at some times decrease and at others increase. The values of *E*,  $\sigma$  and *w*f were taken such that they overall about mimic the observations, but were

not identical to them. These are, as mentioned above, just case studies to illustrate how the various parameters interplay.

p9 l27-28: "Only when the load had reached 2 kPa, all cracks fully propagated towards the end of the column. This finding suggests that the slab was initially not strong enough to support the propagation". I don't understand the logic link between these two sentences (load/strength ?). Clarify.

We suggest that the tensile strength of the slab was initially not large enough so that cracks did not propagate to the very end of the column, but slab failures occurred. Slab density generally increases with increasing load, and tensile strength also increases with density. We will clarify this in the revised manuscript.

p10 I7 "5.9 cm". This is not a range.

Thanks; we will change range to difference.

p10 I15-19: "The errors associated with the parameters ... the dots in the PTV analysis)."This a new info that belongs to Methods and Results sections.

We think it is common practice to discuss errors and uncertainties in the Discussion section, but we will re-consider where to best put this information in the revised manuscript.

p10 I19-22: Adding error-bars on the figures 2a, 3a would help to illustrate this discussion. Moreover, you might go further in this discussion. Indeed w\_f depends only onone layer whereas E is an integrated value on the slab layers and might thus be less sensitive to the spatial variations of one layer.

We will indicate these typical errors in Figures 2a and 3a. Thanks for the suggestion; we think the difference in reproducibility is not related to spatial variations.

p10 l26: "validated" -> "evaluated"

Reuter et al. (2015) in fact validated their SMP-derived metrics with independent observations. Hence we prefer to keep validated.

p11 I3: "is in line with the observations in particular when considering the CT and ECT scores.". What are the others ?

We refer to avalanche activity and will do so explicitly in the revised manuscript.

p11 I10: "- suggesting that the propagation propensity decreased". Delete

We agree that this statement is redundant, but we think it helps the reader to digest the message.

p11 I10-11: "This behavior follows from the fact that two of the essential variables, the bulk modulus and the weak layer shear strength also increase with time." From your sensitivity analysis (figures 6a,b) and the fact that you get the same results for Eq. (1), this is not a sufficient explanation.

We explain in detail in the following lines why we think that  $r_c^{SNP}$  shows this behavior. We will further clarify this in the revised manuscript.

p11 I14-15: "However, it seems premature to rate this metric as it has to be considered as being still in an experimental state." I agree this is a very valuable criterion to help to synthesize the data of snowpack models. However, the explanation is evasive. To my opinion, evaluation of this metric on one point stability observations with potential errors in meteorological forcing and SNOWPACK modeling is the main problem. See main comments. Delete or reword.

We are not aware of any more appropriate way of validating parameters derived from modelled snow stratigraphy other than with measurements in study plots surrounding an automatic weather station. We strongly believe that snow instability predictions from a numerical snow cover model need to be validated with fracture mechanical experiments, or in-situ snow instability tests in general, directly at the location of the weather station. The model of course needs to be driven with these local data otherwise there is already an unknown spatial bias.

p11 I19: "The parameter most strongly influencing the critical cut length seems to be the load". Not shown in results. Can be quantified. See main comments.

We agree that this statement is not supported since we missed to previously mention this in the Results and Discussion sections. We will refer to this result earlier in the revised manuscript.

Figures: what is the running median smoother (kernel size?)

It is a running median with window size 3.

Figure 1: a) give r\_c in m for consistency. b) indicate in the figure what is the black solid line.

The black solid line is described in the figure caption: load as provided by SNOWPACK.

**References**

- Gaume, J., van Herwijnen, A., Chambon, G., Wever, N. and Schweizer, J., 2016. Snow fracture in relation to slab avalanche release: critical state for the onset of crack propagation. Cryosphere Discuss.: in review.
- Gauthier, D., Brown, C. and Jamieson, B., 2010. Modeling strength and stability in storm snow for slab avalanche forecasting. Cold Reg. Sci. Technol., 62(2-3): 107-118.
- Giraud, G., 1993. MEPRA: an expert system for avalanche risk forecasting, Proceedings International Snow Science Workshop, Breckenridge, Colorado, U.S.A., 4-8 October 1992. Colorado Avalanche Information Center, Denver CO, USA, pp. 97-106.
- Heierli, J., Gumbsch, P. and Zaiser, M., 2008. Anticrack nucleation as triggering mechanism for snow slab avalanches. Science, 321(5886): 240-243.
- Jamieson, J.B. and Johnston, C.D., 2001. Evaluation of the shear frame test for weak snowpack layers. Ann. Glaciol., 32: 59-68.
- Jamieson, J.B., Zeidler, A. and Brown, C., 2007. Explanation and limitations of study plot stability indices for forecasting dry snow slab avalanches in surrounding terrain. Cold Reg. Sci. Technol., 50(1-3): 23-34.
- Reiweger, I., Schweizer, J., Ernst, R. and Dual, J., 2010. Load-controlled shear apparatus for snow. Cold Reg. Sci. Technol., 62(2-3): 119-125.
- Reuter, B., Proksch, M., Löwe, H., van Herwijnen, A. and Schweizer, J., 2013. On how to measure snow mechanical properties relevant to slab avalanche release. In: F. Naaim-Bouvet, Y. Durand and R. Lambert (Editors), Proceedings ISSW 2013.

International Snow Science Workshop, Grenoble, France, 7-11 October 2013. ANENA, IRSTEA, Météo-France, Grenoble, France, pp. 7-11.

- Reuter, B., Schweizer, J. and van Herwijnen, A., 2015. A process-based approach to estimate point snow instability. Cryosphere, 9: 837-847.
- Scapozza, C., 2004. Entwicklung eines dichte- und temperaturabhängigen Stoffgesetzes zur Beschreibung des visko-elastischen Verhaltens von Schnee. Ph.D. Thesis, ETH Zurich, Zurich, Switzerland, 250 pp.
- van Herwijnen, A., Gaume, J., Bair, E.H., Reuter, B., Birkeland, K.W. and Schweizer, J., 2016. Estimating the effective elastic modulus and specific fracture energy of snowpack layers from field experiments. J. Glaciol.: in press.
- Vionnet, V., Brun, E., Morin, S., Boone, A., Faroux, S., Le Moigne, P., Martin, E. and Willemet, J.M., 2012. The detailed snowpack scheme Crocus and its implementation in SURFEX v7.2. Geosci. Model Dev., 5(3): 773-791.

---

## Author Comment (AC2) · 29 Jul 2016

**Reply to Referee #2**

We thank the reviewer for the constructive review and valuable comments that will be very helpful for preparing an improved revised manuscript. In the following we reply to the comments in detail and describe the changes we intend to make in the revised manuscript.

This paper presents a unique dataset of temporal changes in crack propagation propensity over the course of a season, and how that propagation propensity related to temporal changes in the slab and weak layer. The authors utilized the latest tools for their work, including analyzing high speed video with PTV, making measurements with the SMP, and modelling the evolution of the snow cover with SNOWPACK. The paper is a valuable addition to the literature, and I believe it should be published after it is revised.

My first suggestion is that the authors consider a different title. Since the paper really focuses on crack propagation propensity, the title should better reflect that. Perhaps something along the lines of "Temporal evolution of crack propagation propensity in view of slab and weak layer properties" or similar? Or, even more specifically, "Temporal evolution of critical cut lengths. . . ??

Thanks for the suggestions. We intend changing the title to "*Temporal evolution of crack propagation propensity in snow in relation to slab and weak layer properties*"

Also, it would be nice if the authors could briefly describe more of the methods used. I know that they will not want to repeat long sections of previous work, but if it would be useful for the reader if they could provide even a few more details about some of the SMP and SNOWPACK derived parameters. More background information will help the reader better assess those parameters and how they performed.

We will describe the methods in more detail so that the paper becomes more self-contained.

**Major comments:**

One primary concern about the paper has to do with Figure 1a and the evolution of the cut length of the PSTs. In this graph it appears that the authors are mixing results that go to END with result that are SFs. Can the authors discuss and defend why they feel this is an appropriate treatment of these data? In my experience I have seen situations where SFs have longer cut lengths, but then as the PSTs transition to END results the cut lengths decrease. At this point, I am not convinced that you can treat the two sets of results (END and SFs) as the same and show a temporal trend with them. I would suggest that they defend this, or they only consider the tests that went to END.

Thanks for rising this point. The critical cut length we report in Figure 1 is independent of the subsequent dynamic phase of crack propagation. Whether or not a crack will arrest, possibly resulting in a slab fracture, or run to the end of the column, will depend on slab as well as weak layer properties – just as the conditions for the onset of the running crack depend on slab as well as weak layer properties. More specifically, recent research indicates that the tensile strength of the slab may decide on how far cracks propagate (Gaume et al., 2015; Schweizer et al., 2014). However, the onset entirely depends on the balance between the energy available for fracture, i.e. the mechanical energy released due an incremental advance of the crack, and the fracture energy, i.e. the energy required for crack growth, or in other words the resistance to crack propagation.

We checked our extensive data base of propagation saw tests and contrast in the figure below the critical lengths for tests with fracture arrests (ARR, SF) and those with full propagation (END) results. There is for our dataset (N = 427) no significant difference between the crack lengths (p = 0.46).

Figure: Critical cut length as observed in PST's as a function of the propagation result. Propagation (on the left) includes all tests where the crack propagated to the very end (END). Arrest (on the right) includes all tests where the crack arrested (ARR) – with or without visible fracture across the slab (SF). Total number of PST results: N = 427, unpublished data.

Furthermore, we certainly agree with the observation by the reviewer that cut lengths may decrease while the result changes from SF to END. In our experience this is usually related to a change in slab properties, e.g. due to additional load, which then will affect the onset as well as the dynamic crack propagation phase. This is actually what we observe towards the end of January 2015 and is likely the reason for the transition from SF to END fractures in our dataset.

Another primary concern about this paper is that I feel a much more robust discussion of the results is warranted. The authors have presented many interesting results, both in terms of the temporal evolution of various parameters and in the comparison of different methods of tracking those temporal changes (between the field tests, PTV, the SMP, and SNOWPACK). However, in my opinion the authors do not fully discuss many of these findings. Some examples:

- The temporal changes of effective elastic modulus of the slab derived from PTV and derived from the SMP do not match (Figure 2). However, this discrepancy is not discussed. Which one of these two techniques do the authors believe is closer to capturing the "true" change in the elastic modulus? It seems to me that the PTV results more closely align with changes I've observed in the field. If this is the case for these data, can the author suggest ways the SMP techniques can be improved?

We agree that there are many open questions, and discrepancies, with regard to the various methods we apply. Most methods have been validated independently, and so far not been contrasted. So far, only Reuter et al. (2013) made an initial attempt to compare various measurements methods; these authors are about to prepare a more in-depth manuscript for a peer-reviewed journal.

We will discuss the discrepancies in the revised manuscript, but it would be beyond the scope to provide a full comparison. What can be said for sure, is that both methods do not provide the true elastic modulus (Young's modulus).

- The SMP's derived critical cut length did not match the observed changes in the PST. Why do the authors think this is the case? Is this some shortcoming in the SMP technique, or are the data presented in this paper somehow different from the data used to develop the SMP derived critical cut length? Does this finding shed additional uncertainty on the SMP derived cut length?

We agree with the reviewer's observation of a certain discrepancy and will discuss how our measurements relate to the validation data presented in Reuter et al. (2015, Figure 8b). The difference in part stems from the fact that the SMP-derived modulus is not really well related to other independent measurements of the modulus (Reuter et al., 2013). We will consider whether it would not be more appropriate to simply derive density from the SMP (Proksch et al., 2015) and then determine the modulus based on the parameterization provided by Scapozza (2004).

- On page 12, line 10 the authors state that this metric is experimental so it is premature to rate it. I disagree with this statement. If the metric is seen as useful enough to be included in the paper, then I feel that it is appropriate to fully evaluate it and rate its usefulness.

We will discuss the issue of the usefulness of the new metric more thoroughly in the revised manuscript. We would like to point out that most validation studies consider many single measurements, compare those to the modelled values and find a correlation, of course including some scatter (e.g., Reuter et al., 2015). Predicting the proper temporal trend seems more challenging, in particular if the trend is rather weak, and changes of mechanical properties from week to week may be small.

- Another point that is not fully discussed is the difference between the elastic modulus values calculated using PTV and those calculated with the SMP. It would be nice to have a paragraph discussing these differences, why they occur, and whether there are ways to get better measurements out of some of the techniques. This could be placed after the paragraph ending on Page 11, Line 19. Looking at Figure 2, the numbers for the SMP seem strange (staying the same or even going down over the season), while the numbers for PTV seem more realistic. What do the authors think about this and how might they explain it?

We agree and will add some more discussion. We have recently shown (van Herwijnen et al., 2016) that the PTV-derived modulus fits relatively well with results from laboratory experiments in the same range of strain rates. For the SMP, the strain rates are presumably higher, yet the SMP-derived modulus is similar – not higher as would be expected due to the higher strain rate. This is likely due to the size of the cone, which is comparable to the grain size in snow, and therefore local effect (force chains, jamming) influence the results.

**Minor comments:**

- Most figures feature a dashed line that is a "running median smoother". It would be helpful to know how the authors calculate this smoother. Also, are the cut lengths in Figure 1a treated the same whether the test went to END or was SF? It appears they were, but the authors may wish to state that in the text. We will specify how running median smoother was calculated; it is a running median with window size 3.

The cut lengths are treated the same whether the crack did run to the end or arrested. See also reply above. We will specify this in the revised manuscript.

- Page 2, Line 1. It is true that Sigrist and Schweizer (2007) were "among the first to emphasize the importance of the slab layers and weak layers", but there were others that emphasized that point either prior to, or at the same time as, 2007. Those include the MS thesis by B.C. Johnson (2001), the paper by Johnson and Jamieson (2004 in CRST), the PhD dissertation by van Herwijnen (2005), the paper by van Herwijnen and Jamieson (2007 in CRST), and the thesis by Gauthier (2007). Since this has been an important point, I'd encourage the authors to add some of those other publications to this citation. Other earlier work also talks about the slab, but in terms of "emphasizing" the slab, it really began to be more clearly stated in the 2000s with the work by Johnson, Jamieson, van Herwijnen, Sigrist and Schweizer.

We will reword the sentence to emphasize the explicit interaction of slab and weak layer properties for evaluating the critical cut length since this is what we refer to. Of course, the general importance of slab properties for crack propagation has been stated previously; we do not mind to add some more references.

However, the studies mentioned by the reviewer primarily focus on the relevance of the slab properties and not explicitly on the interplay between slab and weak layer properties. Only the analysis by Sigrist (2006) finally made it clear how slab and weak layer properties interact in crack propagation, which had already been shown for failure initiation. van Herwijnen and Jamieson (2007) related snowpack properties to failure initiation and crack propagation propensity and showed conceptually how slab properties, in particular slab depth, affect failure initiation and crack propagation in different ways.

- Page 2, line 18 and 19. Do the authors believe that the "shear strength of the weak layer is important for failure initiation" in the case of a triggered avalanche from flat terrain?

Yes, it is certainly important. In flat terrain, a skier not only induces compressive stresses, but also shear stress of similar intensity (Monti et al., 2016; Schweizer, 1997). Hence, the shear stress induced by a skier is very significant even in flat terrain. It is thus more likely that under these mixed-mode conditions the failure begins in shear (or mixed mode) rather than in pure compression because weak layers are weaker in shear than in compression (Reiweger and Schweizer, 2010).

- Page 7, line 6. When the authors state that "cracks did not always fully propagate", it would be useful if they stated how many tests were done and how many propagated (i.e., something along the lines of "when we did the first PSTs, cracks fully propagated in two of five tests, while slab fractures occurred in the other three tests" or whatever the numbers were).

We will add this information as suggested, see also Figure 1a.

- Page 7, line 8. Like above, it would be nicer to know the number of tests instead of just writing "all tests".

We will add this information as suggested.

- Page 7, line 18. It seems to me that the data demonstrate that the propagation propensity decreased definitively (rather than slightly) between the first two days because on the second day all of the tests were SF while on the first day there were some that went to END. This could be due to the shallower nature of the snow in that part of the plot, as discussed by the authors, or it might have to do with a change in the slab. I have observed a decrease in propagation propensity (from more END results to more SF results) when a slab loses tensile strength due to near-surface faceting.

Thanks for pointing this out. We will include this point into the Discussion section.

- Page 9, line 15 and Figure 5. Did all of the ECTs propagate across the column (ECTPs) or did some not (ECTNs)? That should be made clear here or on Figure 5.

Not all ECTs propagated across the column. We will add this information as suggested.

**Typographical/grammatical errors:**

- Page 1, Line 8, add "(PSTs)" after "propagation saw tests" since "PST" is used later in the abstract.
- Page 1, Line 21, delete "considering the slab,"
- Page 1, Line 32, replace "but" with "and"
- Page 2, line 14 and 15. It seems the information for those two sentences comes from a single reference, but the authors cite both Jamieson and Schweizer, 2000 and Schweizer et al., 1998. Which reference is correct?
- Page 3, line 3, delete "exists"
- Page 4, line 13, add an "s" to "PST" so it reads "PSTs"
- Page 5, line 3, delete the comma that is after "modulus"
- Page 7, line 9. You cannot have two semicolons in the same sentence. You will need to re-word to remove one of them.

Thanks for these suggestions which we will consider in the revised manuscript.

**References**

- Gaume, J., van Herwijnen, A., Chambon, G., Birkeland, K.W. and Schweizer, J., 2015. Modeling of crack propagation in weak snowpack layers using the discrete element method. Cryosphere, 9: 1915-1932.
- Monti, F., Gaume, J., van Herwijnen, A. and Schweizer, J., 2016. Snow instability evaluation: calculating the skier-induced stress in a multi-layered snowpack. Nat. Hazards Earth Syst. Sci., 16(3): 775-788.
- Proksch, M., Löwe, H. and Schneebeli, M., 2015. Density, specific surface area and correlation length of snow measured by high-resolution penetrometry. J. Geophys. Res., 120(2): 346-362.
- Reiweger, I. and Schweizer, J., 2010. Failure of a layer of buried surface hoar. Geophys. Res. Lett., 37: L24501.
- Reuter, B., Proksch, M., Löwe, H., van Herwijnen, A. and Schweizer, J., 2013. On how to measure snow mechanical properties relevant to slab avalanche release. In: F. Naaim-Bouvet, Y. Durand and R. Lambert (Editors), Proceedings ISSW 2013.

International Snow Science Workshop, Grenoble, France, 7-11 October 2013. ANENA, IRSTEA, Météo-France, Grenoble, France, pp. 7-11.

- Reuter, B., Schweizer, J. and van Herwijnen, A., 2015. A process-based approach to estimate point snow instability. Cryosphere, 9: 837-847.
- Scapozza, C., 2004. Entwicklung eines dichte- und temperaturabhängigen Stoffgesetzes zur Beschreibung des visko-elastischen Verhaltens von Schnee. Ph.D. Thesis, ETH Zurich, Zurich, Switzerland, 250 pp.
- Schweizer, J., 1997. Contribution on the skier stability index. Internal report, 712, Swiss Federal Institute for Snow and Avalanche Research, Davos, Switzerland.
- Schweizer, J., Reuter, B., van Herwijnen, A., Jamieson, J.B. and Gauthier, D., 2014. On how the tensile strength of the slab affects crack propagation propensity. In: P. Haegeli (Editor), Proceedings ISSW 2014. International Snow Science Workshop, Banff, Alberta, Canada, 29 September 3 October 2014, pp. 164-168.
- Sigrist, C., 2006. Measurement of fracture mechanical properties of snow and application to dry snow slab avalanche release. Ph.D. Thesis, ETH Zurich, Zurich, Switzerland, 139 pp.
- van Herwijnen, A., Gaume, J., Bair, E.H., Reuter, B., Birkeland, K.W. and Schweizer, J., 2016. Estimating the effective elastic modulus and specific fracture energy of snowpack layers from field experiments. J. Glaciol.: in press.
- van Herwijnen, A. and Jamieson, J.B., 2007. Snowpack properties associated with fracture initiation and propagation resulting in skier-triggered dry snow slab avalanches. Cold Reg. Sci. Technol., 50(1-3): 13-22.

---

## Author Response (AR1)

**Swiss Federal Research Institute WSL**
Eidg. Forschungsanstalt WSL
Institut fédéral de recherches WSL
Istituto federale di ricerca WSL

The Cryosphere

Davos, 9 September 2016

WSL Institute for Snow and Avalanche Research SLF

[Figure]

[Figure]

**Submission of revised manuscript**

Dear Editors,

we would like to submit the revised manuscript entitled

"Temporal evolution of crack propagation propensity in snow in relation to slab and weak layer properties"

by Schweizer, Reuter, van Herwijnen, Richter and Gaume.

We thank the reviewers for the helpful comments, which we have all considered while preparing the revised manuscript.
The major changes we made are as follows.

1. We now provide substantially more details on the various Methods. Furthermore, we now use the adjusted mechanical energy as described by van Herwjinen et al. (2016); due to this change we had to re-do the entire PTV and SMP analysis.

2. We now discuss the results more in depth; the Discussion section is much expanded.

3. We have thoroughly revised the manuscript with regard to wording and terminology.

4. We now provide additional profile data in the Supplementary Material and added a new figure.

With these significant changes we hope that our manuscript now meets the quality standards required for publication in The Cryosphere.

Best regards,
Jürg Schweizer
(on behalf of all authors)

WSL Institute for Snow and Avalanche Research SLF
Flüelastrasse 11, CH-7260 Davos Dorf, Switzerland, phone +41 81 417 0111, fax +41 81 417 0110, www.slf.ch

**Reply to Referee #1**

We thank referee #1 for the thorough review. The comments were very helpful for preparing the revised manuscript. In the following, we reply to the comments in detail and describe the changes we made in the revised manuscript.

*SUMMARY:*
*The authors monitored the temporal evolution of a weak layer-slab system during winter 2014-2015 in a field site located next to Davos. Typically, each week between 6 January 2015 and 3 March 2015 (8 days of measurements), they performed on the same site located next to an automatic weather station:*
- *three propagation saw test (PST) on which they measured the critical crack length, the full or partial crack propagation and the slab displacement field (PIV measurements),*
- *around five SMP profiles,*
- *a classical manual snow profile with a density profile*
- *CT/ECT tests.*

*The authors try to explain the observed temporal evolution of the PST critical crack length (general increase with a minimum the 28 January) by investigating the evolution of individual mechanical parameters of the weak layer and slab, namely the load on the weak layer, the weak layer fracture energy and the so-called bulk elastic modulus; and their interaction through the anti-crack model. They used previously developed methods to access these parameters from the measured data. They also used the SNOWPACK model to compute the critical length from the simulated snow profile with meteorological forcings from the automatic weather station. The authors show that monitoring the evolution of individual parameters cannot explain the observed critical crack length trend but that it is necessary to account for the complex interaction between these mechanical variables. The SMP metric is not able to reproduce the observed critical crack length. The SNOWPACK metric shows also an increase of the critical crack length.*

*GENERAL COMMENTS*
*The dataset collected by the authors is very interesting combining quantitative stability analysis (PST critical crack length) and highly resolved vertical hardness profile (SMP). Some of the results are of clear interest to the snow and avalanche community: the authors showed that both slab properties and weak layer cannot be individually monitored to understand the crack propagation propensity evolution; they also show that the previously developed SMP stability metric is not capable of capturing the evolution of the critical crack length. However, the methods are not well presented and appear as a black boxes where explanations on the basic assumptions are missing and the methods are mixed without an apparent logic. In particular, the SMP stability metric presentation is not clear in this form. Evaluating the stability metric of SNOWPACK from a modeled snow profile without showing that the modeled snowpack profile has something in common with the observations is not informative. The sensitivity analysis on a three parameters analytic function is based on four single cases. The trend analysis gives too much importance to a single day case that might be not statistically representative. Therefore, I recommend major revisions before publication.*

- We agree that our description of the methods was minimal mainly referring to previous work. We changed this approach and now provide more details on each of the methods we use.
- We now provide information on the SNOWPACK simulations so that the reader can assess whether the simulated stratigraphy has something in common with the observations (see below).
- What we called sensitivity analysis should be considered as examples of how the critical cut length changes as a function of time for various scenarios of temporal evolution.

Hence, this paragraph was meant to illustrate how the various parameters interact. We changed the title to «Case studies» as it is obviously not a sensitivity analysis.
- With regard to the temporal evolution and the observed minimal values towards the end of January, we now discuss the representativity more thoroughly. We would like to point out that minimal values obtained with snow instability tests are in general more trustworthy. In the case of the propagation saw test, any measurement and observation errors increase the cut length. Low values of the cut length therefore almost always represent the real conditions.

*MAJOR COMMENTS:*

*1) The dataset collected by the authors is very valuable. Indeed, the authors present it as the first comprehensive time series of a weak layer, slab system. It uses state-of-the-art measuring techniques (SMP, PST) combined with "traditional" measurements (manual stratigraphy and density, CT/ECT). Since one of the objective and strength of the paper is this dataset, it appears logical to provide this dataset as supplementary files (Caaml file for stratigraphy, stability tests, text file for SMP and avi file for PST videos).*

We now provide the manual profiles as well as the SMP profile performed at the profile location for each day as Supplementary Material.
Providing further data is not straightforward. We are not dealing with 'simple' weather data, but with data from various sources (SMP, PTV, SNOWPACK, manual snow profiles), which then have to be processed to get to the results. Furthermore, the processing is not trivial.
In addition, we included a figure (new Figure 1) to the main text showing the SNOWPACK simulation (see below), and the manual snow profile as well as the SMP profile for one specific date (28 January 2015).
And of course, we will provide the data on request to others who like to collaborate.

[Figure]

Figure: (a) SNOWPACK simulation for the location of the automatic weather station (AWS) WAN7 for winter 2014-2015 showing the evolution of grain shape, black vertical line indicates date of snow profile (28 Jan 2015), (b) simulated snow profile for 28 Jan 2015, (c) manually observed snow profile at the location of the AWS on 28 Jan 2015, (d) corresponding SMP penetration force signal measured at the location of the manual profile. Red arrows point to the weak layer.

*2) The writing style on the mechanical background is often unscientific and requires precision and consistency. I have listed some of these problems:*
- *about the elastic modulus. You used the following terms without proper definition:"elastic modulus", "bulk modulus", "modulus", "effective modulus", "bulk effective modulus", "mi-*

*cro-mechanical modulus", "slab modulus", "stiffness", "elastic modulus with non-elastic parts of deformation". This vocabulary is misleading and is not suited for a scientific paper, where the mechanical concepts behind the used model should be precisely presented, which can be done in a simple way accessible to the snow community.*

We carefully revised the manuscript to increase precision and consistency.
For example, we now consistently use the term «effective elastic modulus», and no longer use the terms «stiffness» and «bulk modulus». However, we still frequently use «modulus» or «slab modulus» when we refer to the elastic properties (of the slab) in general.

We acknowledge that the vocabulary on the deformation behavior of the slab was hard to follow. Part of the problem arises from the fact that model assumptions, e.g. linear elasticity, do not fit what is actually observed and can be measured in the field; in addition, the slab is layered and not uniform. Therefore, there is some need for specific terms and it is not sufficient to just talk about the modulus. For example, the modelling approach by Heierli et al. (2008) includes the elastic modulus (Young's modulus), what we measure with PTV is an effective bulk modulus (bulk because layering is disregarded, effective because it includes not only purely elastic parts of deformation), what is derived from the SMP is the micro-mechanical modulus. Nevertheless, as pointed out above, we now consistently use the term «effective elastic modulus» whenever feasible.

  - *you use the terms "propagation propensity", "propagation criterion r_c_SMP" ,"critical crack length", "propagation propensity metric", "crack propagation propensity" to refer to the same parameter r_c, or maybe not but this is not clear. Why don't you use consistently the well-defined "critical crack length" and explain only in the introduction that the critical crack length is an indicator of the more general concept of crack propagation propensity?*
  - *"initiation probability", "initiation propensity", "initiation criterion", "initiation indices", "skier stability index" ...*
  - *delete vague and unspecific claims "reliable", "reliable in general", "distinct pattern", "relevant mechanical properties", "other mechanical properties"*

We thoroughly went through the manuscript and removed redundant or confusing terms, in particular in connection with failure initiation and crack propagation. However, as we present measured as well as modelled values there is some need for distinction between the various measures.
In addition, we now provide definitions of crack propagation propensity and snow instability (page 3, lines 19-27).
Moreover, we thoroughly went through the manuscript and removed vague and unspecific terms such as distinct.

*3) It is hard to follow the history of the weak layer-slab system. It is necessary to add a one-page figure with eight sub-figures (one for each day of measurements) showing the manual stratigraphy (at least snow type and density), a SMP profile and the position of the weak layer.*

We now provide a figure (new Figure 1 in the revised manuscript, see above) showing the SNOWPACK simulation as well as modelled and observed profiles for 28 January 2015 (see above). In addition, we provide all the manually observed profiles including an SMP profile in the Supplementary Material.

*4) In Heierli's model, the total mechanical energy of a PST crack of length r is composed of two terms: V(r) = w_f \* r + Vm(r) where w_f \* r is the weak layer fracture energy and Vm(r)*

*accounts for elastic deformation energy and changes in gravity potential energy of the slab. In case of a uniform slab, Vm(r) can be computed analytically knowing the density, thickness and elastic modulus of the slab. In case of a FE model of a multilayer slab (density, thickness and elastic modulus per layer known), Vm(r) can be calculated numerically. This is done for the SMP analysis. In case of a measured displacement/deformation field of the PST tests, Vm(r) can also be calculated.*

*This is done in the PST analysis. In both cases (SMP, PST method), the calculated Vm(r) is used to fit the analytic mono-layer solution. The fitted analytic solution is then differentiated to obtain the critical crack length knowing the weak layer fracture energy (SMP method) or the weak layer fracture energy knowing the critical crack length(PST method). I don't understand why the dVm(r)/dr is not computed directly from the calculated Vm(r)(or with smoothing of Vm(r)). This is not explained in the proposed references (Reuter et al, 2015 or van Herwijnen and Heierli, 2010). The bulk elastic modulus is a fitting parameter and it is unclear how physically-relevant it is. There is no clear reason why Vm(r) on layered material should fit directly the mono-layer analytic solution. Provide a proper explanation and discussion on that. Moreover, recall the main hypothesis (elastic linear, only the slab contributes to deformation energy) of Heierli's model.*

We now provide more details in the Methods section and also refer to the recent paper by van Herwijnen et al. (2016) where the PTV method is explained in detail. We reanalyzed all data and now use their refined approach, i.e. the adjusted mechanical energy to account for differences between the model of Heierli et al. (2008) and the FE simulations.

Taking the derivative of the raw data to derive $w_f$ would not work, as there is too much scatter and this would result in very unreliable values of $w_f$.

We agree that the critical cut length can be computed with the FE model using the SMP slab properties and the SMP-derived specific fracture energy $w_f$, but would require an iterative approach to find $r_c$.

*5) Section 2.4 describing the SMP signal processing is vague and unscientific. Many critical details are missing. It does not allow the reader to reproduce the presented method and appears as a black box. It requires a deep rewriting. It mixes method using different concepts that measures the same things differently e.g. Johnson and Schneebeli (1999) and shot-noise model used by Proksch, 2015. The window size for analysis, the SMP version, the adjustment parameters of (Proksch et al, 2015, calculated on a few alpine snow samples), the finite element layer mesh, etc. are missing.*

*There is additional linear scaling with no convincing explanation. The calculation of layer Young's modulus from SMP elementary failure element is known to be poor and is inconsistent with the one based on density (Scapozza, 2004) used by the snow cover modeling (p5 l30). The failure initiation criterion S is not detailed and it is hard to notice that it does not incorporate snow load in comparison to SK38 which does, ... The reference to other papers is far from being sufficient and clear explanations won't take more than 30 lines.*

As mentioned above we now provide more details in general and in particular on the methods and not simply refer to previous work.

The additional linear scaling is simply introduced to obtain SMP-derived values that are comparable to other macroscopic mechanical properties since the raw processed data only represent microscopic values not directly related to common material properties.

In the absence of a sound calibration of the SMP-derived microstructural properties scaling with the PTV-derived values represents a reasonable alternative. This is now more clearly described and discussed (page 7, lines 17-22; page 16, lines 6-11).

Furthermore, it is clear that SMP-derived values have some deficiencies, see Reuter et al. (2013). We now discuss this more thoroughly in the revised manuscript (page 15, lines 31-33; page 16, lines 1-5). In addition, we included an alternative approach for determining the

effective elastic modulus by using the SMP-derived density and the density-modulus relation reported by Scapozza (2004) (page 7, lines 23-26).

*6) The authors used the snow cover model forced by a nearby automatic weather station as an input of a new critical crack length estimator (Gaume et al. 2014a, 2016). Without any clue on how close the snowpack simulation to the observed snowpack, it is impossible to exploit the results of this analysis. It is well-known that one point evaluation of a snow cover model on stability criterion is difficult. Note that the only variables missing in Eq. (1) is the weak layer strength that could be fitted to get r_c_snp = r_c_obs, similarly to what is done for the PST.*

*Additionally, it is not clear to me how the avalanche activity index (concerning the area all around Davos?) can help to analyze the measurement done in this particular site.*

The SNOWPACK simulation reproduced the snow stratigraphy reasonably well – with the notable exception that the melt-freeze crust (resulting from a high-elevation rain event) below the weak layer was not simulated. We now provide the simulated stratigraphy (new Figure 1 in the revised manuscript; see above).
We certainly agree that stability predictions from simulated snow stratigraphy are challenging. We strongly believe that these stability predictions should be validated at locations of automatic weather stations.
With regard to the comment on Eq. (1), we agree that the only missing variable is the weak layer strength, however, we are not sure we understand the reviewer's point. The shear strength cannot be determined form the measured critical cut length, otherwise the model would no longer be predictive. The shear strength is obtained from the parametrization implemented in the snow cover model SNOWPACK based on the work of Jamieson and Johnston (2001).
As we perform our measurements in a representative study plot commonly used in operational forecasting to extrapolate to the surrounding terrain (e.g., Gauthier et al., 2010; Jamieson et al., 2007), we added the avalanche activity data for comparison with the local stability evaluations. This is not better motivated (page 9, lines 20-22).

*7) The pattern of the PST critical crack length is a general increase with a local minimum for one measurement day (28 January). As discussed (p6 l20-23, p10 l3-6), the spatial variability can significantly affect the stability even a few meters away. Given the poor representativity of one day of measurement to define a trend, and potential spatial variability, it would be reasonable when speaking of trend to not focus on the minimum observed the 28 January but on the general trend (continuous increase ofr_c). Note that this does not challenge the fact that the SMP should reproduce the same trend (since measured a few cm away from the PST); but the comparison with SNOWPACK is challenged. The explanations "we deem it unlikely that the observed pattern is entirely the result of spatial variability and does not reflect the temporal evolution", "Previous studies performed in level study plots have shown that measurements in general are reliable and that the effect of spatial variations is relatively small" are not convincing, at least in this form.*

We re-considered the local minimum that we observed at the end of January 2015. In fact, low critical cut lengths were not only observed on 28 January but also on 5 February. On 5 February there are only two measurements with a large difference between them. However, low critical cut lengths are in general more trustworthy than high ones, if they concurrently occur, since any error while performing the test will increase the cut length. Furthermore, on 28 January 2015, for the first time, all cracks propagated to the end of the PST column indicating that the crack propagation propensity had increased. Finally, the additional loading towards the end of January 2015 resulted in many avalanches and shooting cracks were

frequently observed also indicating increased propagation propensity. We re-assessed the issue of measurement accuracy and spatial variability, reworded the corresponding statements (page 14, lines 11-33).

The sentence "*Previous studies performed in level study plots have shown that measurements in general are reliable and that the effect of spatial variations is relatively small*" is supported by two references to previous work just following this sentence (page 14, lines 27-30).

*8) The sensitivity analysis is poor and based on four different cases. To my opinion, this cannot be called a sensitivity analysis. Differentiating Eq. (2) with respect to E, sigma and wf provides a way to perform this sensitivity analysis properly.*

*Note that the general comments are general and require re-wording of several parts of the paper and additional explanations, and not only taking into account specific minor points listed below.*

The purpose of this paragraph is to illustrate how changes of the modulus, the load and the specific fracture energy with time will affect the temporal evolution of the critical cut length. We no longer call this a sensitivity study, but selected a more appropriate title for the paragraph: «Case studies».

We agree, that differentiating Eq. (2) with respect to $E$, $\sigma$ and $w_f$ would reveal the dependence of the critical cut length for a single parameter. However, these dependencies, considered independently are obvious: the cut length decreases with increasing load, and increases with increasing slab modulus and weak layer fracture energy. However, their interplay in course of time cannot easily be assessed – and the four examples we provide simply show that entirely different evolutions are possible.

*MINOR COMMENTS:*

*abstract: the following terms are too vague : "distinct pattern", "other mechanical properties" "some of the relevant mechanical properties"*

We removed "distinct" throughout the manuscript, and clarified the terms: " *… by simply monitoring mechanical properties such as slab load, slab modulus or weak layer specific fracture energy.*"

*p1 l25: "how much stress due to a skier is transferred". Misleading sentence. All the stress is transferred to the ground. But it is distributed on a larger surface. Reword.*

We reworded the sentence: "*… the slab layers determine the magnitude of the stress due to a skier at the depth of the weak layer*" (page 1, line 29).

*p1 l28: "with respect to the weak layer, a snowpack a weakness is" -> "the weak layer is"*

We reworded the sentence as suggested (page 2, line 2).

*p2 l2: "conceptual model". Describe this model in a few words.*

We added a sentence describing the effect of slab thickness (or weak layer depth): "*With increasing slab depth conditions for failure initiation become less favourable whereas conditions for crack propagation become more favourable.*" (page 2, lines 10-11).

*p2 l7: "though the strengthening may lag behind the loading". Sound unscientific. Delete.*

We reworded to: *"… the strength increase may lag behind the loading during a snowfall."* (page 2, line 15).

*p2, l27: References to the model Surfex-Crocus (Vionnet, V. et al. Model Development The detailed snowpack scheme Crocus and its implementation in SURFEX v7.2. Geoscientific Model Development 5, 773–791 (2012)) and Mepra (e.g. 1. Giraud, G.MEPRA an expert system for avalanche risk forecasting. in International Snow Science Workshop 97–104 (1992)) are clearly missing.*

Thanks for pointing this out. We are certainly aware of the French model forecasting chain and rate it highly. We added two references to articles mentioning MEPRA (Giraud, 1993; Vernay et al., 2015) (page 3, line 3)
However, it is unclear to us how the temporal evolution of strength is modeled, and in general how the strength is derived. We are not aware that in the various publications about MEPRA this is described in detail.

*p3 Section 2.1: Is the snowpack completely dry during measurement period?*

Yes, the snowpack was completely dry – apart from some melting at the surface in early January resulting in a thin crust (see new Figure 1).

*p4 l1-2: "The weak layer . . . December 2014". Explain how you know that.*

We know as we closely follow the snowpack evolution and are in the field several times a week. This was the decisive weak layer at the end of December 2014. As mentioned on page 4, lines 16-19, there are no profiles available that were performed at fracture lines to support this assumption, but the particular weak layer consistently showed up as the primary failure layer in snow instability tests in the days following the avalanche cycle.

*p4 l2-3: "While no fracture . . . January 2015". I don't understand. Reword.*

See reply above. We reworded the last three sentences of this paragraph (page 4, lines 20-23).

*p4 l7: "The manual snow profile served as a reference". Do you mean that you performed manual stratigraphic matching to adjust the other snow profiles to the manual profile?*

The manual snow profile served as a reference to, for example, indicate the depth of the weak layer or other prominent layers.

*p4 l10: "at least three PST". It appears from Figure 1a) that there two other dates where less PST were performed.*

As mentioned some test results had to be discarded since the cut was not performed consistently close to the interface which we only realized once we analyzed the videos. For that reason, we only have two test results on two days (21 January and 3 February 2015). We now provide this information (page 5, lines 9-11).

*p4 l14: "we cut the layer of faceted crystals at its upper interface". One of the main difficulty of the PST is to follow the weak layer of interest. As explained in Section2.1, there was another FC layer just above the weak layer of interest. Showing the SMP profiles (see main comments) could help the reader to evaluate the likelihood of deviation of the saw cut in the weak layer.*

As we filmed all tests we can easily assess whether the tests were properly performed – and have of course done so (see reply above). We now also discuss the difficulties or properly performing the tests (page 14, lines 14-18).

As mentioned above, we will provide one SMP profile per measurement day in the Supplementary Material. However, the SMP profiles are less suited to assess a potential deviation while cutting the weak layer.

*p4 l18: Give version of SMP.*

We used SMP version 2 (page 5, line 20).

*p4 l25: "the displacement of the markers was used to estimate the mechanical energy Vm (r) with increasing crack length". As far as I understand, at this step, you also need the load, i.e. the density of the manual profile. Add explanation if this is correct.*

Thanks for pointing this out; we added that the density of the manual profile is used to evaluate the mechanical energy (page 6, line 18).

*p4 section 2.3: The critical crack length of the modeled PST is inherently equal (or very close) to the observed critical crack length since the observation is used to fit w_f. This might not appear clearly to the reader. Please add this kind of explanation.*

The critical crack length is modelled from the weak layer fracture energy $w_f$ as derived from the SMP. It is independent of the observed critical crack length. We now better explain the derivation of the modelled critical cut length $r_c^{SMP}$ in section 2.4 and explicitly mention that the SMP-derived modelled critical cut length is independent of the observed critical cut length (page 8, lines 10-11).

*p5 l28: "the shear modulus of the weak layer which was estimated". How ?*

Following Gaume et al. (2016) we used a constant value of the shear modulus $G_{WL}$ = 0.5 MPa according to previous results of laboratory experiments by Reiweger et al. (2010) and Camponovo and Schweizer (2001); for the Poisson's ratio of the slab we assumed a value of 0.2.
We reworded this statement to: *"For the shear modulus of the weak layer we assumed a constant value of 0.5 MPa, based on laboratory experiments (Camponovo and Schweizer, 2001; Reiweger et al., 2010)."* (page 9, lines 12-14).

*p5 l30: I suggest to explicitly indicate the power law relation used here.*

We now provide the relation as suggested (page 7, line 26).

*p6 Eq2: To my opinion, this equation in this form does not give any information to the reader. Delete or give detail on all terms.*

We now introduce the equation earlier in the Methods section and provide all details (now Eq. 7 in the revised manuscript).

*p6 l23-26: "By then, the weak layer of ... resulting in a load of almost 4 kPa." Belong to the load section 3.2?*

This part of the sentence simply makes the link between slab thickness and density on one hand and load on the other hand so that the reader can better relate load values to commonly used parameters such as slab thickness.

*p7 l29: "0.3 J m -2 to about 1.5 J m -2". Recall that this range results from a linear scaling between w_f_SMP and w_f_PTV.*

We now recall the scaling in the Discussion section (page 16, lines 6-11).

*p8 l3: S = shear_strength / skier stress should be described in Methods. Adding two lines of description is not a big deal and would clarify the message. See main comments.*

As mentioned above, we now introduce the SMP-derived metrics of instability in more detail in the Methods section of the revised manuscript.

*p8 l10: SK38 = shear_strength / (skier stress + weight_stress) should be described in Methods. See main comments.*

We now introduce the SK38 in the Methods section of the revised manuscript (page 9, lines 1-4).

*p8 l22-24: The CT/ECT tests could be better used to evaluate the initiation criteria (SMP, SK38).*

Thanks for this suggestion; we now discuss the CT/ECT results with respect to the initiation criteria in the revised manuscript (page 16, lines 31-32; page 17, lines1-4; page 17, line 19).

*p9 l14-18: I don't understand this paragraph. The rc_obs is used to compute w_f_PTV. That w_f_PTV as input in Heierli's model gives the same trend for r_c does not appear to me as a finding ??? Clarify.*

This paragraph is to illustrate that under certain assumptions for the temporal evolution of $E$, $\sigma$ and $w_f$ the critical cut length can at some times decrease and at others increase. The values of $E$, $\sigma$ and $w_f$ were taken such that they overall about mimic the observations, but were not identical to them. These are, as mentioned above, just case studies to illustrate how the various parameters interplay.

*p9 l27-28: "Only when the load had reached 2 kPa, all cracks fully propagated towards the end of the column. This finding suggests that the slab was initially not strong enough to support the propagation". I don't understand the logic link between these two sentences (load/strength ?). Clarify.*

We suggest that the tensile strength of the slab was initially not large enough so that cracks did not propagate to the very end of the column, but slab failures occurred. Slab density generally increases with increasing load, and tensile strength also increases with density. We now better explain this in the revised manuscript (page 13, lines 24-26).

*p10 l7 "5.9 cm". This is not a range.*

We now specify that the range is the difference between minimal and maximal values (page 14, line 19).

*p10 l15-19: "The errors associated with the parameters ... the dots in the PTV analysis)."This a new info that belongs to Methods and Results sections.*

We think it is common practice to discuss errors and uncertainties in the Discussion section.

*p10 l19-22: Adding error-bars on the figures 2a, 3a would help to illustrate this discussion. Moreover, you might go further in this discussion. Indeed w_f depends only on one layer*

*whereas E is an integrated value on the slab layers and might thus be less sensitive to the spatial variations of one layer.*

We now show the errors in Figures 3a and 4a. Thanks for the suggestion; we added this point to the Discussion section (page 16, lines 18-20).

*p10 l26: "validated" -> "evaluated"*

Reuter et al. (2015) in fact validated their SMP-derived metrics with independent observations. Hence we prefer to keep validated.

*p11 l3: "is in line with the observations in particular when considering the CT and ECT scores.". What are the others ?*

We referred to avalanche activity.

*p11 l10: "– suggesting that the propagation propensity decreased". Delete*

We have completely re-worded this paragraph.

*p11 l10-11: "This behavior follows from the fact that two of the essential variables, the bulk modulus and the weak layer shear strength also increase with time." From your sensitivity analysis (figures 6a,b) and the fact that you get the same results for Eq. (1), this is not a sufficient explanation.*

We explain in detail in the following lines why we think that $r_c^{SNP}$ shows this behavior. We tried to further clarify this in the revised manuscript (page 17, lines 29-34; page18, lines 1-12).

*p11 l14-15: "However, it seems premature to rate this metric as it has to be considered as being still in an experimental state." I agree this is a very valuable criterion to help to synthesize the data of snowpack models. However, the explanation is evasive. To my opinion, evaluation of this metric on one point stability observations with potential errors in meteorological forcing and SNOWPACK modeling is the main problem. See main comments. Delete or reword.*

We are not aware of any more appropriate way of validating parameters derived from modelled snow stratigraphy other than with measurements in study plots surrounding an automatic weather station. We strongly believe that snow instability predictions from a numerical snow cover model need to be validated with fracture mechanical experiments, or in-situ snow instability tests in general, directly at the location of the weather station. The model of course needs to be driven with these local data otherwise there is already an unknown spatial bias.

*p11 l19: "The parameter most strongly influencing the critical cut length seems to be the load". Not shown in results. Can be quantified. See main comments.*

We agree that this statement is not supported since we missed to previously mention this in the Results and Discussion sections. We now discuss this finding earlier in the revised manuscript.

*Figures: what is the running median smoother (kernel size?)*

The dashed lines now simply connect the median values per day.

*Figure 1: a) give r_c in m for consistency. b) indicate in the figure what is the black solid line.*

The black solid line is described in the figure caption: load as provided by SNOWPACK.

***Reply to Referee #2***

We thank the reviewer for the constructive review and valuable comments that were very helpful for preparing the revised manuscript. In the following, we reply to the comments in detail and describe the changes we made in the revised manuscript.

*This paper presents a unique dataset of temporal changes in crack propagation propensity over the course of a season, and how that propagation propensity related to temporal changes in the slab and weak layer. The authors utilized the latest tools for their work, including analyzing high speed video with PTV, making measurements with the SMP, and modelling the evolution of the snow cover with SNOWPACK. The paper is a valuable addition to the literature, and I believe it should be published after it is revised.*

*My first suggestion is that the authors consider a different title. Since the paper really focuses on crack propagation propensity, the title should better reflect that. Perhaps something along the lines of "Temporal evolution of crack propagation propensity in view of slab and weak layer properties" or similar? Or, even more specifically, "Temporal evolution of critical cut lengths. . ."?*

Thanks for the suggestion. We changed the title to «*Temporal evolution of crack propagation propensity in snow in relation to slab and weak layer properties*».

*Also, it would be nice if the authors could briefly describe more of the methods used. I know that they will not want to repeat long sections of previous work, but if it would be useful for the reader if they could provide even a few more details about some of the SMP and SNOWPACK derived parameters. More background information will help the reader better assess those parameters and how they performed.*

We now describe the methods in more detail so that the paper becomes more self-contained.

*Major comments:*
*One primary concern about the paper has to do with Figure 1a and the evolution of the cut length of the PSTs. In this graph it appears that the authors are mixing results that go to END with result that are SFs. Can the authors discuss and defend why they feel this is an appropriate treatment of these data? In my experience I have seen situations where SFs have longer cut lengths, but then as the PSTs transition to END results the cut lengths decrease. At this point, I am not convinced that you can treat the two sets of results (END and SFs) as the same and show a temporal trend with them. I would suggest that they defend this, or they only consider the tests that went to END.*

Thanks for rising this point. The critical cut length we reported in Figure 1a (now Figure 2a) is independent of the subsequent dynamic phase of crack propagation. Whether or not a crack will arrest, possibly resulting in a slab fracture, or run to the end of the column, will depend on slab as well as weak layer properties – just as the conditions for the onset of the running crack depend on slab as well as weak layer properties. More specifically, recent research indicates that the tensile strength of the slab may decide on how far cracks propagate (Gaume et al., 2015; Schweizer et al., 2014). However, the onset entirely depends on the balance between the energy available for fracture, i.e. the mechanical energy released due an incremental advance of the crack, and the fracture energy, i.e. the energy required for crack growth, or in other words the resistance to crack propagation (see page 5, lines 14-19).

We checked our extensive data base of propagation saw tests and contrast in the figure below the critical lengths for tests with fracture arrests (ARR, SF) and those with full propagation (END) results. There is for our dataset ($N$ = 427) no significant difference between the crack lengths ($p$ = 0.46).

[Figure]

Figure: Critical cut length as observed in PST's as a function of the propagation result. Propagation (on the left) includes all tests where the crack propagated to the very end (END). Arrest (on the right) includes all tests where the crack arrested (ARR) – with or without visible fracture across the slab (SF). Total number of PST results: $N$ = 427, unpublished data.

Furthermore, we certainly agree with the observation by the reviewer that cut lengths may decrease while the result changes from SF to END. In our experience this is usually related to a change in slab properties, e.g. due to additional load, which then will affect the onset as well as the dynamic crack propagation phase. This is actually what we observe towards the end of January 2015 and is likely the reason for the transition from SF to END fractures in our dataset.

*Another primary concern about this paper is that I feel a much more robust discussion of the results is warranted. The authors have presented many interesting results, both in terms of the temporal evolution of various parameters and in the comparison of different methods of tracking those temporal changes (between the field tests, PTV, the SMP, and SNOWPACK). However, in my opinion the authors do not fully discuss many of these findings. Some examples:*

- *The temporal changes of effective elastic modulus of the slab derived from PTV and derived from the SMP do not match (Figure 2). However, this discrepancy is not discussed. Which one of these two techniques do the authors believe is closer to capturing the "true" change in the elastic modulus? It seems to me that the PTV results more closely align with changes I've observed in the field. If this is the case for these data, can the author suggest ways the SMP techniques can be improved?*

We agree that there are some open questions, and discrepancies, with regard to the various methods we apply. Most methods have been validated independently, and so far not been contrasted. So far, only Reuter et al. (2013) made an initial attempt to compare various measurements methods; these authors are about to prepare a more in-depth manuscript for a peer-reviewed journal.

We now discuss the discrepancies in more detail, but it is beyond the scope of the manuscript to provide a full comparison of the methods. This also requires a much larger dataset.

- *The SMP's derived critical cut length did not match the observed changes in the PST. Why do the authors think this is the case? Is this some shortcoming in the SMP technique, or are the data presented in this paper somehow different from the data used to develop the SMP derived critical cut length? Does this finding shed additional uncertainty on the SMP derived cut length?*

We agree with the reviewer's observation of a certain discrepancy. We have re-analyzed the data based on the new findings by van Herwijnen et al. (2016) and the temporal evolution of the SMP-derived critical cut length does now better match.
The difference in part stems from the fact that the SMP-derived modulus is not really well related to other independent measurements of the modulus (Reuter et al., 2013). We now provide an alternative approach by deriving density from the SMP (Proksch et al., 2015) and then determining the modulus based on the parameterization provided by Scapozza (2004).

- *On page 12, line 10 the authors state that this metric is experimental so it is premature to rate it. I disagree with this statement. If the metric is seen as useful enough to be included in the paper, then I feel that it is appropriate to fully evaluate it and rate its usefulness.*

We agree and now discuss the results for the new metrics in substantially more detail.

- *Another point that is not fully discussed is the difference between the elastic modulus values calculated using PTV and those calculated with the SMP. It would be nice to have a paragraph discussing these differences, why they occur, and whether there are ways to get better measurements out of some of the techniques. This could be placed after the paragraph ending on Page 11, Line 19. Looking at Figure 2, the numbers for the SMP seem strange (staying the same or even going down over the season), while the numbers for PTV seem more realistic. What do the authors think about this and how might they explain it?*

We agree and now discuss this issue in more detail. We have recently shown (van Herwijnen et al., 2016) that the PTV-derived modulus fits relatively well with results from laboratory experiments in the same range of strain rates. For the SMP-derived modulus this calibration however is lacking.

*Minor comments:*
- *Most figures feature a dashed line that is a "running median smoother". It would be helpful to know how the authors calculate this smoother. Also, are the cut lengths in Figure 1a treated the same whether the test went to END or was SF? It appears they were, but the authors may wish to state that in the text.*

We have replaced the median smoother. The dashed lines we now show, simply connect the median values per day.
The cut lengths are treated the same whether the crack did run to the end or arrested (resulting in a slab fracture). See also reply above. We now explicitly mention this in the Methods section of the revised manuscript (page 5, lines 14-19).

- *Page 2, Line 1. It is true that Sigrist and Schweizer (2007) were "among the first to emphasize the importance of the slab layers and weak layers", but there were others that*

*emphasized that point either prior to, or at the same time as, 2007. Those include the MS thesis by B.C. Johnson (2001), the paper by Johnson and Jamieson (2004 in CRST), the PhD dissertation by van Herwijnen (2005), the paper by van Herwijnen and Jamieson (2007 in CRST), and the thesis by Gauthier (2007). Since this has been an important point, I'd encourage the authors to add some of those other publications to this citation. Other earlier work also talks about the slab, but in terms of "emphasizing" the slab, it really began to be more clearly stated in the 2000s with the work by Johnson, Jamieson, van Herwijnen, Sigrist and Schweizer.*

We reworded the sentence to emphasize the explicit interaction of slab and weak layer properties for evaluating the critical cut length: *"Sigrist and Schweizer (2007) first described the interaction of slab and weak layer properties for evaluating the critical cut length. By interpreting their results in a fracture mechanical framework they concluded that the energy that has to be exceeded to fracture a weak layer depends on the material properties of the weak layer, whereas the energy that is available for crack propagation mainly depends on the material properties of the overlaying slab, and may also depend on the collapse height of the weak layer."* (page 2, lines 3-7)

- *Page 2, line 18 and 19. Do the authors believe that the "shear strength of the weak layer is important for failure initiation" in the case of a triggered avalanche from flat terrain?*

Yes, it is certainly important. In flat terrain, a skier not only induces compressive stresses, but also shear stress of similar intensity (Monti et al., 2016; Schweizer, 1997). Hence, the shear stress induced by a skier is very significant even in flat terrain. It is thus more likely that under these mixed-mode conditions the failure begins in shear (or mixed mode) rather than in pure compression because weak layers are weaker in shear than in compression (Reiweger and Schweizer, 2010).

- *Page 7, line 6. When the authors state that "cracks did not always fully propagate", it would be useful if they stated how many tests were done and how many propagated (i.e., something along the lines of "when we did the first PSTs, cracks fully propagated in two of five tests, while slab fractures occurred in the other three tests" or whatever the numbers were).*

We now provide this information in the revised Table 1, and it can also be seen in Figure 2a.

- *Page 7, line 8. Like above, it would be nicer to know the number of tests instead of just writing "all tests".*

We added the number of tests as suggested (page 19, lines 7-9).

- *Page 7, line 18. It seems to me that the data demonstrate that the propagation propensity decreased definitively (rather than slightly) between the first two days because on the second day all of the tests were SF while on the first day there were some that went to END. This could be due to the shallower nature of the snow in that part of the plot, as discussed by the authors, or it might have to do with a change in the slab. I have observed a decrease in propagation propensity (from more END results to more SF results) when a slab loses tensile strength due to near-surface faceting.*

Thanks for pointing this out. We included this point into the Discussion section.

- *Page 9, line 15 and Figure 5. Did all of the ECTs propagate across the column (ECTPs) or did some not (ECTNs)? That should be made clear here or on Figure 5.*

Almost all ECTs propagated across the column. We now provide this information in the revised Table 1.

*Typographical/grammatical errors:*

Thanks for these suggestions which we considered in the revised manuscript as follows:

- *Page 1, Line 8, add "(PSTs)" after "propagation saw tests" since "PST" is used later in the abstract.*

Added as suggested.

- *Page 1, Line 21, delete "considering the slab,"*

Deleted as suggested.

- *Page 1, Line 32, replace "but" with "and"*

Changed as suggested.

- *Page 2, line 14 and 15. It seems the information for those two sentences comes from a single reference, but the authors cite both Jamieson and Schweizer, 2000 and Schweizer et al., 1998. Which reference is correct?*

Whereas the study by Jamieson and Schweizer (2000) is more comprehensive, the strength increase of the order 100 Pa d$^{-1}$ is only mentioned in Schweizer et al. (1998).

- *Page 3, line 3, delete "exists"*

We prefer to keep "exists".

- *Page 4, line 13, add an "s" to "PST" so it reads "PSTs"*

Changed as suggested.

- *Page 5, line 3, delete the comma that is after "modulus"*

Deleted as suggested.

- *Page 7, line 9. You cannot have two semicolons in the same sentence. You will need to re-word to remove one of them.*

Removed as suggested.

Schweizer, J., 1997. Contribution on the skier stability index. Internal report, 712, Swiss Federal Institute for Snow and Avalanche Research, Davos, Switzerland.

[revised manuscript text omitted]

---

## Author Response (AR3)

**Swiss Federal Research Institute WSL**
Eidg. Forschungsanstalt WSL
Institut fédéral de recherches WSL
Istituto federale di ricerca WSL

The Cryosphere

Davos, 19 October 2016

WSL Institute for Snow and Avalanche Research SLF

[Figure]

[Figure]

**Submission of revised manuscript**

Dear Editors,

we would like to submit the revised manuscript entitled

"Temporal evolution of crack propagation propensity in snow in relation to slab and weak layer properties"

by Schweizer, Reuter, van Herwijnen, Richter and Gaume.

We thank the Editor for the last two comments which we have fully considered while preparing the revised manuscript.
With these clarifications we hope that our manuscript now meets the quality standards required for publication in The Cryosphere.

Best regards,
Jürg Schweizer
(on behalf of all authors)

WSL Institute for Snow and Avalanche Research SLF
Flüelastrasse 11, CH-7260 Davos Dorf, Switzerland, phone +41 81 417 0111, fax +41 81 417 0110, www.slf.ch

**Reply to comments by Editor**

We thank the Editor for his final two comments on the revised manuscripts. We have fully considered them.

Page 6, line 5: As suggested, we now define density as well as slope angle.

Page 16, line 1: We now already recall in the first paragraph that the SMP values were scaled and refer to the following paragraph. We hope that with these additions the reasoning becomes clearer. In fact, even though the SMP-values are scaled, they do not really agree in our specific case. The reason stems from the fact that contrasting SMP- and PTV-derived values (to derive the scaling) reveals that there is a lot of scatter so that specific values can be quite a bit off.

[revised manuscript text omitted]